# Improving Pluvial Flood Simulations with Multi-source DEM Super-Resolution

Yue Zhu[1,2], Paolo Burlando[1], Puay Yok Tan[3], Christian Geiß[4,5], and Simone Fatichi[6]

[1]Institute of Environmental Engineering, ETH Zurich, Switzerland
[2]Future Cities Laboratory, Singapore-ETH Centre, Singapore
[3]Department of Architecture, National University of Singapore, Singapore
[4]German Remote Sensing Data Center (DFD), German Aerospace Center (DLR), Germany
[5]Department of Geography, University of Bonn, Germany
[6]Department of Civil and Environmental Engineering, National University of Singapore, Singapore

*Correspondence to*: Yue Zhu (yue.zhu@sec.ethz.ch)

**Abstract.** Accurate flood simulation remains a significant challenge in many flood-prone regions, particularly in developing countries and urban areas, where the availability of high-resolution topographic data is especially limited. While publicly available Digital Elevation Model (DEM) datasets are increasingly accessible, their spatial resolution is often insufficient for reflecting fine-scaled elevation details, which hinders the ability to simulate pluvial floods in built environments. To address

this issue, we implemented a deep learning-based method, which efficiently enhances the spatial resolution of DEM data, and quantified the effect of the improved DEM on flood simulation. The method employs a tailored multi-source input module, enabling it to effectively integrate and learn from diverse data sources. By utilizing publicly open global datasets, low-resolution DEM datasets (i.e., 30m SRTM) in conjunction with high-resolution multispectral imagery (e.g., Sentinel-2A), our approach allows to produce a super-resolution DEM, which exhibits superior performance compared to conventional methods

in reconstructing 10m DEM data based on 30m DEM data and 10m multispectral satellite images. We evaluated the performance of the super-resolution DEM in flood simulations. Compared to conventional methods (e.g., bicubic interpolation), the simulation results demonstrated that our approach significantly improved the accuracy of flood simulations, with a reduction in the mean absolute error of floodwater depth of about 13.1% and an increase in the IoU for inundation area predictions of about 46%. Accordingly, this study underscores the practical value of machine-learning techniques that leverage

publicly available global datasets to generate DEMs that allow enhancing flood simulations.

## 1 Introduction

The occurrence of severe urban floods has been on the rise, partly influenced by climate change, which contributes to more frequent extreme rainfall events (Tabari, 2020). To address these challenges, high-resolution flood modelling is essential for making informed flood management decisions (Sanders et al., 2024; Y. Wang et al., 2018). As one of the

key inputs for flood simulations, accurate Digital Elevation Model (DEM) data supports reliable flood simulation, in

turn enabling the assessment of various flood mitigation strategies. However, the fidelity of flood simulations is heavily contingent upon the spatial resolution of DEM data (Hawker et al., 2018). At present, open datasets of DEM data with global coverage are predominantly available at raster resolutions coarser than (or equal to) 30m (Marsh et al., 2023), failing to capture the fine-resolution local topography details that are crucial for flood modelling (Hawker et al., 2018).

The lack of publicly open high-resolution DEM is particularly affecting data-scarce regions of the developing world, which are often the most vulnerable to the devastating impacts of floods (Malgwi et al., 2020). In response to these challenges, this research examines the effect of implementing a deep learning-based image super-resolution model for generating high-resolution DEM data and demonstrates the performance of the enhanced DEM data on improving pluvial flood simulations.

**1.1 Existing methods for improving the spatial resolution of DEM**

Methods to enhance the spatial resolution of DEM data have been widely adopted across geospatial applications to improve risk estimates. These advancements have significantly enhanced the accuracy and reliability of natural hazard mapping, including flood prediction (Löwe & Arnbjerg-Nielsen, 2020; Tan et al., 2024), landslide modelling (Brock et al., 2020), volcanic flow assessment (Deng et al., 2019), and snow avalanche forecasting (Miller et al., 2022). The most

widely adopted approaches in existing studies of DEM super-resolution can be categorised into interpolation-based, data fusion-based, and learning-based methods (Zhou et al., 2023). The interpolation methods, such as bilinear and bicubic interpolations (Rees, 2000), are based on the concept of spatial autocorrelation, which posits that points in closer proximity are more alike than those that are more distant (Arun, 2013). While being straightforward and computationally efficient, their performance is often limited by the simplicity of terrain continuity and smoothness, potentially leading to

over-smoothed terrain features (Y. Zhang & Yu, 2022). Data fusion-based approaches combine the strengths of data from different sources to create a more accurate and comprehensive representation of terrain (Yue et al., 2015). During the fusion process, tools such as elevation error maps or weight maps are commonly used to assign importance to each DEM source, ensuring that higher-quality data has a greater influence on the final output. However, these methods often introduce inaccuracies by altering elevation values and failing to address edge effects (Okolie & Smit, 2022), such as

abrupt transitions or mismatches between overlapping DEM datasets.

Deep learning methods have significantly advanced the field of single-image super-resolution, achieving superior performance in reconstructing high-resolution images from their low-resolution counterparts (Yang et al., 2019). The implementations of deep learning-based super-resolution methods have been shown to substantially improve the performance of remote sensing applications (Ling & Foody, 2019; Shang et al., 2022; Xie et al., 2022) and promote the

utilisation of spatial data that was previously underutilised due to limited spatial resolution (Zhu et al., 2021), including the applications of enhancing low-resolution DEM data (Demiray et al., 2021a, 2021b; Jiang et al., 2023; Kubade et al.,

2020; Z. Li et al., 2023; Yue et al., 2015; Zhou et al., 2023, 2021, 2021). For instance, Demiray et al. (2021) utilized Generative Adversarial Networks (GANs) to upscale low-resolution DEMs (50ft) to high-resolution DEMs (3ft), although this study demonstrated the potential of adversarial training in spatial resolution enhancement, GANs are known for instability in training, facing challenges such as mode collapse and vanishing gradients (Jabbar et al., 2021). Zhou et al. (2021) introduced a double-filter deep residual neural network, leveraging residual learning to improve feature extraction and enhance the accuracy of reconstructed DEMs. More recently, Li et al. (2023) proposed a transformer-based deep learning network for upscaling DEM across multiple upsampling factors (e.g., ×2, ×4), showcasing the effectiveness of attention mechanisms in capturing long-range dependencies and spatial relationships. Building on the advances of these existing methods, we refine a DEM super-resolution method by employing a computationally efficient architecture with attention mechanisms to achieve accuracy and robustness. In the previous studies, the majority of the deep learning applications in DEM super-resolution only employed low-resolution DEM data as input, without incorporating additional information. This is a limitation to accurately capture the unique terrain features that characterise high-resolution DEM (Zhou et al., 2023) and that are required to support accurate flood modelling.

## 1.2 Multi-source deep learning for remote sensing applications

In general remote sensing applications, the benefits of integrating multi-source inputs have been increasingly recognised, as the combination of complementary data sources enhances the robustness and reliability of model performance (J. Li et al., 2022). For instance, Shen et al. (2019) developed a deep learning-based model for drought monitoring, which employed multi-source remote sensing data as input, including DEM data, and meteorological and soil data. Lu et al. (2022) proposed a deep learning framework taking Google Earth imagery and point of interest heatmap as input data for urban functional zone extraction. Blöschl et al. (2024) integrated riverbed geometry information into the DEM to enhance national-scale flood hazard mapping.

With respect to the input for DEM super-resolution, it can be argued that, solely relying on a single source of low-resolution (LR) DEM input can be an ill-posed task, as high-resolution details can hardly be accurately reconstructed without additional reference information (Yue et al., 2016). Studies have been made to include additional features generated from low-resolution DEM data. For instance, Zhang et al. (2023) calculated terrain gradient maps based on DEM data to guide the optimisation process of a Convolutional Neural Network (CNN)-based DEM super-resolution. Zhou et al. (2023) proposed a terrain feature-based CNN for DEM super-resolution, which extracts slope and aspect from low-resolution DEM data and deploys them as additional features for model inputs and loss functions.

Besides generating additional features based on low-resolution DEM, efforts have also been made to fuse different data sources to offer fine-granular details related to terrain features, which can improve performance. One example following this direction is found in Argudo et al. (2018), who examined the feasibility of combining natural colour aerial

images together with low-resolution DEM data as input to train a CNN for producing high-resolution DEM, suggesting improved performance compared with interpolation-based methods. Tan et al. (2024) introduced a deep learning-based DEM upscaling network that uses high-resolution optical images to predict elevation differences, and then fuses these predictions with the original DEM data through additional convolutional layers. It should be noted that these studies mainly employed natural colour images for feature fusion. In contrast, multispectral images can provide further features from non-visible wavelengths, such as near-infrared, allowing for more detailed and specialised analysis. This is supported by Chen et al. (2013), showcasing the effects of utilising multispectral bands of satellite images on improving the performance of an interpolation-based DEM densification method. More recently, a few attempts have explored the effects of integrating low-resolution DEM with remote sensing imagery for DEM super-resolution. Gao & Yue (2024) used the red band of Sentinel-2 images to provide auxiliary high-frequency information for DEM super-resolution training. Paul & Gupta (2024) incorporated 3-band satellite images with low-resolution DEM to develop a GAN-based DEM super-resolution model.

## 1.2 Significance of this study

In this literature context, this study aims to investigate the effectiveness and quantification of how pluvial flood simulations can be improved by using a deep learning-based DEM super-resolution construction method, which incorporates multispectral imagery, including the near-infrared band, as additional input. Accordingly, we develop an integrated methodological framework that allows for enhancing input data quality for practical improvements in flood simulation performance, specifically quantifying the extent to which the proposed method of DEM resolution enhancement can contribute to improved pluvial flood hazard simulations. Specifically, we provide the following main contributions: (i) we develop an efficient DEM super-resolution method that incorporates a tailored input module for processing multi-source and multi-scale input data, including both low-resolution DEM data and high-resolution multispectral satellite images; (ii) by using publicly open datasets we ensure the generalizability of the method, especially for DEM-related applications in data-scarce regions; (iii) we provide a quantitative assessment of the performance of the generated super-resolution DEM maps with regards to pluvial flood simulations. The latter is achieved by evaluating the flood inundation maps generated based on different DEM super-resolution methods in terms of both floodwater depth and inundated area. Overall, this study represents a methodological advancement that showcases the practical value of multi-sourced deep learning-based methods for enhancing pluvial flood simulations, thus offering an exemplary pathway to address the issue of lacking high-resolution DEM for reliable risk assessments in the context of land use planning and disaster management.

**2 Methodology**

To improve the spatial resolution of DEM data for enhancing flood simulations, we further develop a deep learning-based DEM super-resolution method. This method employs the Residual Channel Attention Network (RCAN) (Y.Zhang et al., 2018) as the backbone structure and incorporates a tailored multi-source input block to leverage multi-sourced input data, contributing to improved performance in reconstructing high-resolution DEM data.

**2.1 Residual Channel Attention Network (RCAN)**

RCAN is a widely recognised method for single-image super-resolution (Y.Zhang et al., 2018). One of the key features employed in an RCAN is a deep residual network structure that integrates residual in residual (RIR) blocks. The RIR block combines long and short skip connections, enabling the network to learn more high-dimensional features from low-resolution to high-resolution images with a very deep structure, meanwhile avoiding the issue of vanishing gradient during training processes. Another key feature of RCAN is its use of channel attention mechanisms within each residual block. The channel attention mechanism weighs the importance of each channel, thus allowing RCAN to adaptively emphasise features from more important channels while suppressing less useful ones, thereby optimising reconstruction performance by focusing on the most significant features. This channel-attention mechanism can be particularly useful for processing multi-source input data (X. Liu et al., 2021). By integrating the RIR structure with channel-attention mechanisms, RCAN can extract and exploit hierarchical features from the input image effectively. However, since RCAN has been developed for image super-resolution tasks on single natural colour images, we tailored the structure of its input module to handle inputs from different data sources.

**2.2 Multi-source and multi-scale input data fusion**

The proposed method, referred to as RCAN-Multispectral (RCAN-MS), incorporates a tailored multi-source and multi-scale input module, which is the key distinction from the original RCAN. The input module enables the integration of high-resolution multispectral satellite images with low-resolution DEM data, leveraging the complementary information from both sources to reconstruct high-resolution DEMs with enhanced accuracy (Fig. 1). Multispectral satellite images contain information captured across various spectral bands, including both visible light and near-infrared bands, which offer a wealth of information about surface materials, vegetation coverage, water bodies, and other landscape features (Carrão et al., 2008), making them ideal for compensating for the coarse information in low-resolution DEMs. By combining high-resolution multispectral imagery with low-resolution elevation data, deep learning models can access a more comprehensive feature set, facilitating the reconstruction of detailed topographic information.

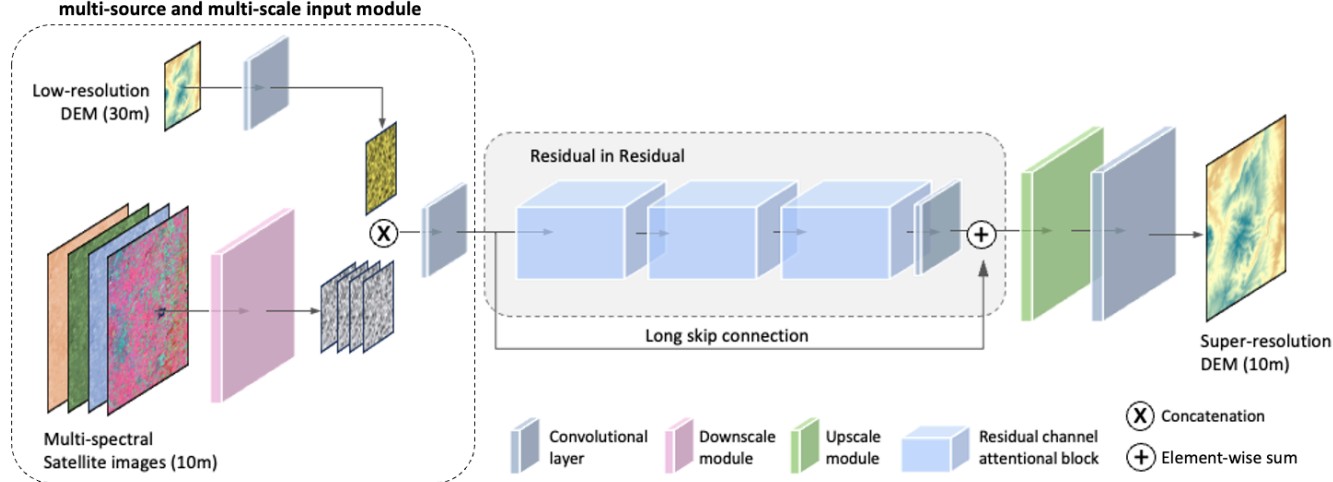

**Fig. 1 The structure of the proposed DEM super-resolution model, MS-RCAN. Low-resolution DEM data and four-band multispectral satellite images are fused using a tailored multi-source and multi-scale input module to facilitate the reconstruction of high-resolution DEM data.**

150    The tailored multi-source input module is integrated into the model structure before the first layer of the RCAN backbone structure (Fig. 1). In particular, the 30m DEM data is fed into a 2D convolutional layer with a kernel size of 3x3, a stride of 1, and a padding size of 1. Meanwhile, the 10m multispectral satellite images are passed to another 2D convolutional layer that has a kernel size of 3x3 but with a stride of 3, which can effectively reduce the spatial dimensions of the input by a factor of 3. A Rectified Linear Unit (ReLU) activation function follows the convolution, introducing

155    non-linearity and enhancing feature representation. As such, the information in the 4-band multispectral input is encoded to a 4-channel tensor that has the same size as the encoded low-resolution data flow. Upon the same size of the two data flows, they can be concatenated along the channel dimension, and then processed by another 2D convolutional layer for data fusion of spatial and spectral information from multi-source inputs. After that, the concatenated multi-source input is passed through the RCAN backbone structure, which consists of RIR blocks and includes a 2D convolutional layer at

160    the end of the model structure to upscale the data flow to the size of the high-resolution DEM map. The proposed method is tested with two datasets at different geographical locations by comparing it with a series of baseline models in the following sections.

## 3 Experiment settings

### 3.1 Datasets

165    Although publicly open DEM datasets with global coverage have limited spatial resolution (approximately 30m or coarser), the spatial resolution of publicly open multispectral satellite imagery with global coverage can reach 10m

spatial resolution, such as Sentinel-2A, which has great potential to provide fine-grained features adding complementary information for DEM super-resolution. Considering the availability of datasets and the generalizability at global scales, we chose to work with a scale factor of ×3 for testing DEM super-resolution models, which is also a scale factor widely adopted in most of the existing studies on image super-resolution (P. Wang et al., 2022).

Two datasets at different geographical locations were employed in this study for training and evaluation of DEM super-resolution methods, as well as for the simulation of pluvial floodwater distribution. The data are all collected from publicly open sources, including SRTM, TanDEM-X, and Sentinel-2, which have been widely adopted for remote sensing applications, e.g., urban environments (Wu, et al., 2019; Geiß et al., 2015; C. Li, et al., 2021). SRTM utilised dual radar antennas to collect interferometric radar data, which was then processed into digital topographic data with a resolution of 1 arcsecond (Farr et al., 2007). TanDEM-X mission uses a single-pass interferometric synthetic aperture radar (InSAR) system to produce 12 m resolution global digital surface models. The Sentinel-2 satellites carry the Multi-Spectral Instrument (MSI), which captures imagery in 13 spectral bands, with the blue, green, red, and near-infrared bands having a 10m spatial resolution (Spoto et al., 2012).

The two selected locations cover the areas of (i) England, UK (Dataset 1), and (ii) Shenzhen and Hong Kong, China (Dataset 2). Each dataset contains three different data sources (Table 1), including a 10m high-resolution DEM map, a 30m low-resolution DEM map, and the corresponding 10m multispectral satellite image composed of four bands (i.e., red, blue, green, near-infrared). It should be noted that, although the spatial resolution of the high-resolution DEM data in both datasets was pre-processed at the same resolution of 10m, they were collected from different sources due to

**Table 1. Information on the DEM data and multispectral satellite images in two datasets for the tests of DEM super-resolution models**

| | | Dataset 1. England | Dataset 2. Shenzhen & Hong Kong |
|---|---|---|---|
| 10m DEM | Collection source | LIDAR Composite DTM 2019, published by UK Environment Agency (2023) | TanDEM-X, provided by German Aerospace Centre (DLR)) |
| | Spatial resolution | Resampled from 2m to 10m resolution using a bilinear interpolation | Resampled from 12m to 10m resolution using a bilinear interpolation |
| | Acquisition date | 2019-09-01 | 2016-01-13 |
| 30m DEM | Collection source | Shuttle Radar Topography Mission (SRTM), published by NASA JPL (2013) | Shuttle Radar Topography Mission (SRTM) , published by NASA JPL (2013) |
| | Spatial resolution | 1 arc-second (~ 30m) resolution | 1 arc-second (~ 30m) resolution |
| | Acquisition date | 2014-09-23 | 2014-09-23 |
| 10m Multispectral Images | Collection source | Sentinel-2A | Sentinel-2A |
| | Spatial resolution | 10m resolution | 10m resolution |
| | Bands | Band 2 – Blue, Band 3 – Green, Band 4 – Red, Band 8 - Near-infrared | Band 2 – Blue, Band 3 – Green, Band 4 – Red, Band 8 - Near-infrared |
| | Acquisition date | 2022-11-25 / 2023-01-21/ 2023-02-13 | 2023-12-25 |

185  data availability. However, such differences in data sources can also be leveraged to test the robustness and generalizability of the proposed methods.

As shown in Fig. 2, each dataset for the test on DEM super-resolution methods was split into three subsets, which are the training set, validation set, and test set. There are no spatial overlapping areas between the three subsets. In each dataset, the three subsets were randomly subsampled into 2000, 200, and 300 small patches for training, validation, and

190  testing, respectively. The size of the subsampled low-resolution DEM patches is 80×80 pixels, and the sizes of the subsampled high-resolution DEM and multispectral images are 240×240 pixels. Correspondingly, the DEM super-resolution models are trained with a target of upscaling the DEM data to three times its original size.

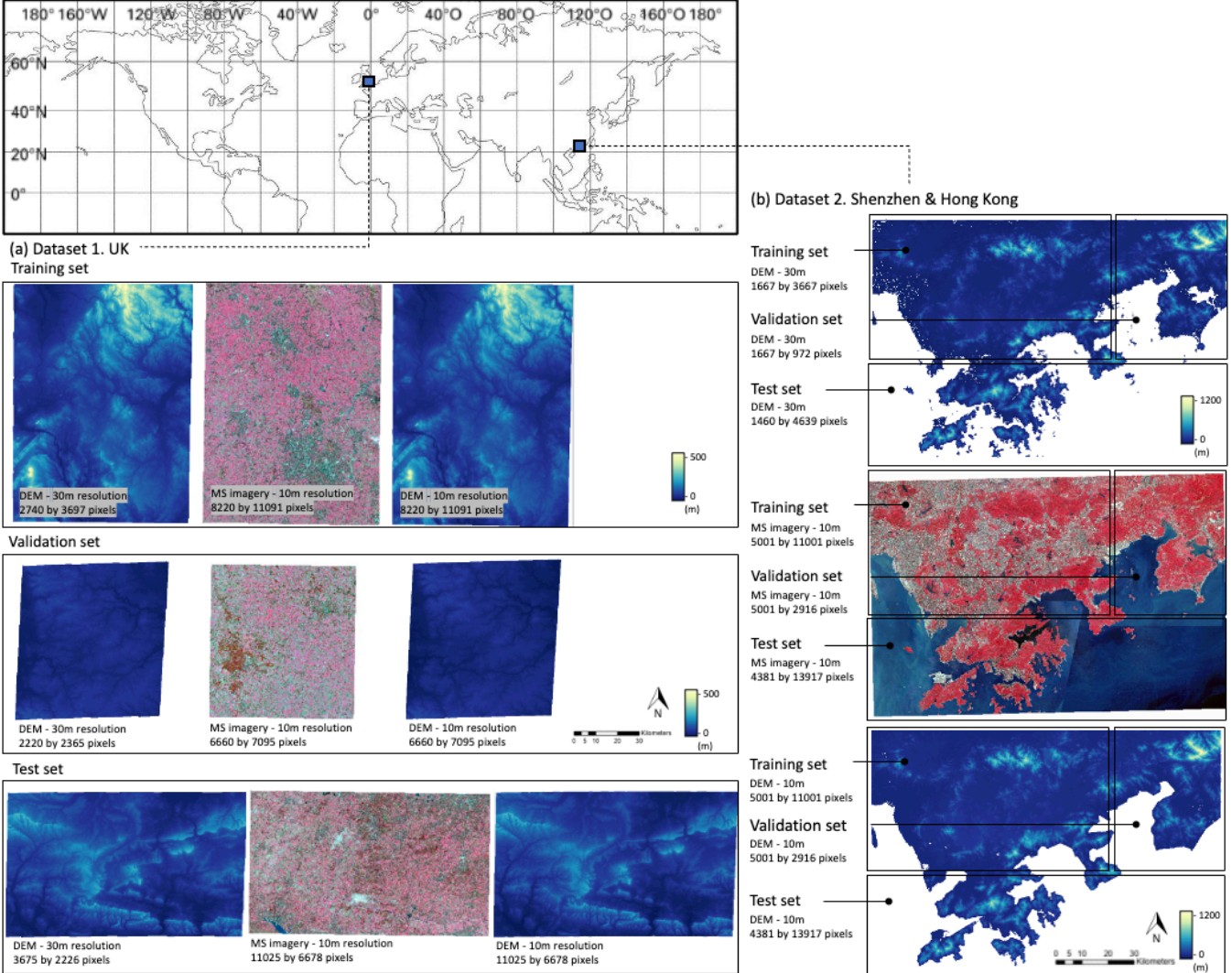

**Fig. 2 Overview of the two datasets for DEM Super-resolution. (a) the training, validation, and test sets of Dataset 1. (b) the training, validation, and test sets of Dataset 2 (see Table 1 for data source).**

## 3.2 Experiment setup

The experiments were composed of two main stages (Fig. 3): (i) DEM super-resolution, and (ii) pluvial flood simulation. The first stage was centred on assessing the performance of DEM Super-resolution methods in enhancing the resolution of the original DEM data, whereas the second stage was to quantify the effects of adopting the super-resolution DEM on enhancing pluvial flood simulations. This quantification offers two main benefits: (i) provides a more comprehensive performance evaluation of how DEMs generated through different methods perform in impact applications; (ii) examines whether the proposed deep-learning approach provides a cost-efficient solution for improving flood simulations.

In the first stage, besides using the original high-resolution DEM data for performance evaluation, four additional baseline methods were employed for comparison with the performance of the proposed method, RCAN-MS. These baseline methods include a conventional bicubic interpolation method, and three other widely adopted neural network-based super-resolution methods, which are Super-Resolution Convolutional Neural Network (SRCNN, Dong et al., 2016), Very Deep Convolutional Network (VDSR, Kim et al., 2016), and Residual Channel Attention Network (RCAN, Y.Zhang et al., 2018). All the models were trained and validated separately in the UK and China datasets, facilitating the evaluation of model performance in learning terrain-specific representations across different geographical contexts.

The training of DEM Super-resolution models was established and trained with PyTorch on two NVIDIA GeForce RTX 4090 GPUs on high-performance computing (HPC) clusters. All baseline models were implemented using the default parameter settings for hidden layers as specified in their original papers. The input and output layer configurations were adapted to suit the task of DEM super-resolution. The baseline methods used as benchmarks utilised single-band input and output layers, except for RCAN-MS, which was configured with five input bands (i.e., single-band DEM and four-band multispectral image). All the test methods adopted the same training strategy, they were all trained with a

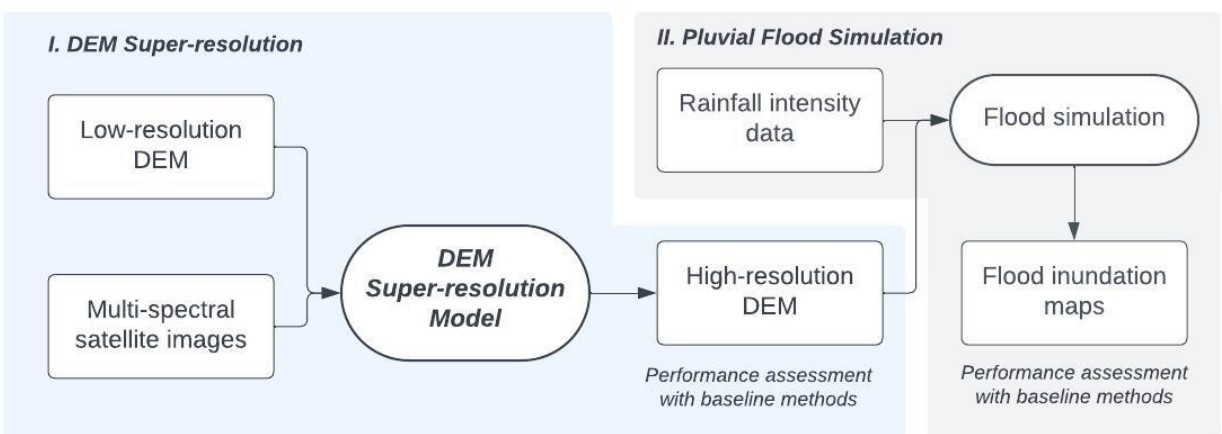

**Fig. 3. Workflow of the experiments, including (i) establishing a DEM Super-resolution model to reconstruct a high-resolution DEM, then (ii) adopting the high-resolution DEM for pluvial flood simulations.**

batch size of 8 and a learning rate of $1\times10^{-4}$. With an adaptive learning rate scheduler, the learning rate decreases to a fraction of 0.8 when the validation loss stops decreasing for 50 epochs. The optimiser adopted for all the methods is Adam with default momentum parameters. The loss function is the Mean Absolute Error (MAE). Regarding the stopping criteria for mode performance evaluation, all the models were trained for 200 epochs with the data in the training set, after which the epoch yielding the smallest MAE values on the validation set was selected for further performance evaluation on the test sets.

The MAE, Mean Square Error (MSE), Peak Signal-to-Noise Ratio (PSNR) and Structural Similarity Index Measure (SSIM) were employed to evaluate the performance of DEM super-resolution models. PSNR and SSIM are two widely used evaluation metrics in image super-resolution tasks (Dong et al., 2016; Kim et al., 2016; Yang et al., 2019). The PSNR measures the quality of reconstruction of a lossy transformation (e.g., image compression) (Z. Wang et al., 2021), with higher values indicating a smaller difference. The SSIM measures the similarity between two images, specifically targeting changes in brightness, contrast, and structure (Z. Wang et al., 2021).

For both the UK and China datasets, the DEM data generated by all the downscaling methods in the super-resolution test were adopted as input for flood simulation. In each dataset, a subarea of 450 by 600 pixels was cropped for pluvial flood simulation. The simulation is conducted using a cellular automata model, Caddies (Guidolin et al., 2016), which is known for efficient pluvial flood simulation in urban environments (H. Liu et al., 2018; Y. Wang et al., 2019; Y. Wang et al., 2023; Zhu et al., 2024).

The pluvial flood simulation was based on a 100-year return period, and 30-minute duration rainfall as a forcing scenario. Since the two geographical locations feature different climate conditions, the rainfall intensity corresponding to the 100-year return period rainfall was set based on the Intensity-Duration-Frequency (IDF) curves for the UK and Hong Kong regions, respectively. The IDF curve for the UK area is computed using the hourly rainfall data recorded at the rain gauge station of Seathwaite, North England, and downloaded from the MIDAS UK sub-hourly rainfall observations dataset (Met Office, 2006). The adopted IDF curve for Hong Kong is according to a report published by the Civil Engineering and Development Department, the Government of the Hong Kong Special Administrative Region (Tang & Cheung, 2011). Specifically, the 100-year return period rainfall intensity for a 30-minute duration is 42 mm/h for Dataset 1, and 190 mm/h for Dataset 2. The simulated flood maps were evaluated by comparing their similarity with the flood map generated based on reference high-resolution DEM data. Both flood depth values and flood area coverage were assessed. The evaluation metrics to measure flood depth values are MAE and MSE, and the metric for evaluating the flood areas is Intersection over Union (IoU), which is a widely adopted method to assess the accuracy of a predicted area in comparison to the target area found in ground truth data (Rahman & Wang, 2016).

# 4 Results

## 4.1 DEM Super-resolution

The experimental results of the comparison between the proposed DEM super-resolution method and the baseline methods are presented in Table 2, including the comparisons in both datasets. For Dataset 1, the RCAN-MS method demonstrates a marked improvement over the Bicubic method, reducing the MAE from 3.0 to 2.2 m (-26.7%), and the MSE from 19.0 to 8.7 m$^2$ (-54.2%). This enhancement is also reflected in the values of PSNR (+9.9%) and SSIM (+34.8%), suggesting a substantially improved fit to the target high-resolution DEM. Similarly, Dataset 2 results reveal

that RCAN-MS significantly outperforms the bicubic interpolation method, with the MAE sharply decreasing from 9.9

**Table 2. Evaluation results of all the tested DEM resolution enhancement methods on two test sets with different geographical locations**

|  | Test set of Dataset 1. | | | | Test set of Dataset 2. | | | |
|---|---|---|---|---|---|---|---|---|
|  | MAE (m) | MSE (m$^2$) | PNSR | SSIM | MAE (m) | MSE (m$^2$) | PNSR | SSIM |
| bicubic | 3.0078 | 19.0206 | 33.4055 | 0.4621 | 9.2924 | 163.0170 | 35.4505 | 0.6091 |
| SRCNN | 2.7665 | 15.5027 | 34.2901 | 0.5776 | 6.8153 | 94.1950 | 37.8500 | 0.6794 |
| VDSR | 2.6530 | 13.4866 | 34.8653 | 0.5737 | 6.6412 | 88.7638 | 38.1110 | 0.6811 |
| RCAN | 2.5967 | 12.9453 | 35.0460 | 0.5975 | 6.4150 | 83.5288 | 38.3950 | 0.6838 |
| RCAN-MS | **2.1952** | **8.7102** | **36.7605** | **0.6205** | **5.8181** | **66.6251** | **39.3543** | **0.7411** |

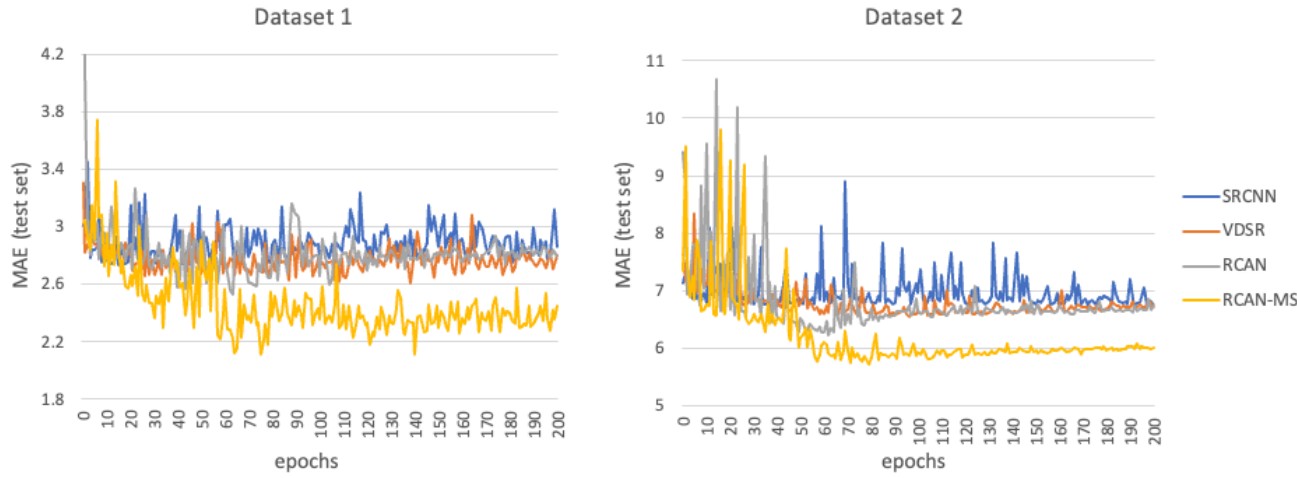

**Fig. 4 Changes in the MAE values of all the tested models as training epochs increase for Dataset 1 (left) and Dataset 2 (right).**

to 5.9 m (-40.4%), and the MSE from 186.0 to 67.6 m$^2$ (-63.7%). The RCAN method, serving as the backbone method for RCAN-MS, shows better results than the other deep learning-based methods such as SRCNN and VDSR across both datasets, underscoring the superior performance of the RCAN-based architecture in the task of DEM resolution enhancement. Specifically, for Dataset 1, RCAN posts an MAE of 2.60 m and an MSE of 12.9 m$^2$, which are better than those for SRCNN and VDSR. In Dataset 2, RCAN achieves an MAE of 6.4 m and an MSE of 83.5 m$^2$, further confirming its robustness. The performance superiority of the proposed RCAN-MS method is evident across all metrics in both datasets, demonstrating its enhanced capability in generating high-fidelity super-resolution DEM data. This is exemplified by the significant reductions in MAE and MSE and the corresponding increase in PNSR and SSIM values.

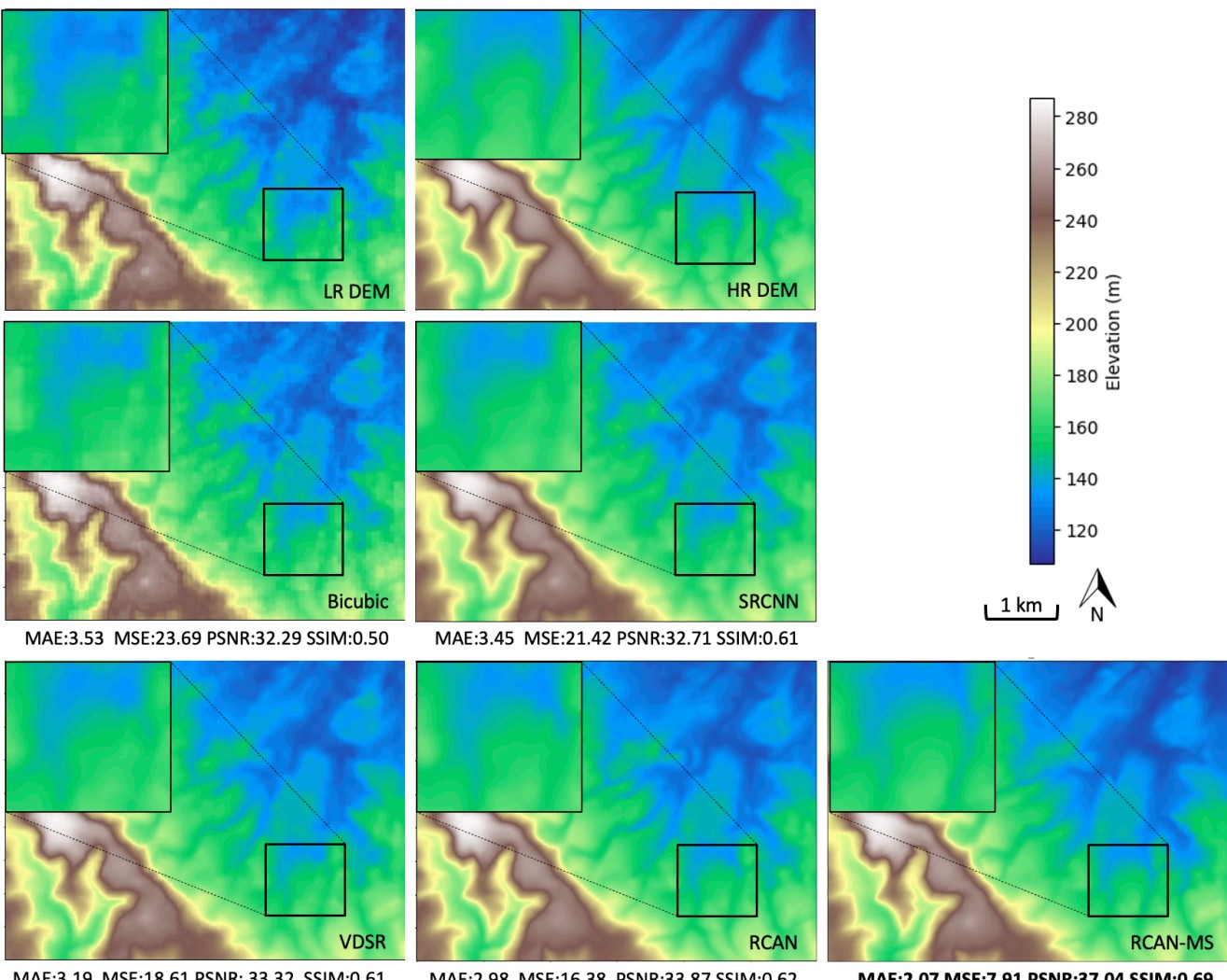

**Fig. 5 Comparison of DEM maps in the test set of Dataset 1. generated by the proposed method, RCAN-MS, other baseline methods and the original high-resolution DEM map in Dataset 1. The values of the evaluation metrics (i.e., MAE, MSR, PSNR, SSIM) for each method are presented.**

Fig. 5 and Fig. 6 present the two selected patches from the test sets of Datasets 1 and 2 for visual assessment of the performance of the super-resolution DEM maps, in which a subarea of exemplary patches is additionally enlarged for further visual comparison of details. The corresponding reference low-resolution DEM and high-resolution DEM maps are also presented for comparison. The ranking of the performance of all the tested methods is aligned with the overall evaluation of the test sets reported in Table 2, suggesting that RCAN offers a larger magnitude of enhancement than SRCNN and VDSR, and RCAN-MS stands out among all the tested methods, recording the lowest MAE and MSE values. These two exemplary patches of the test sets are employed for pluvial flood simulation in the following section.

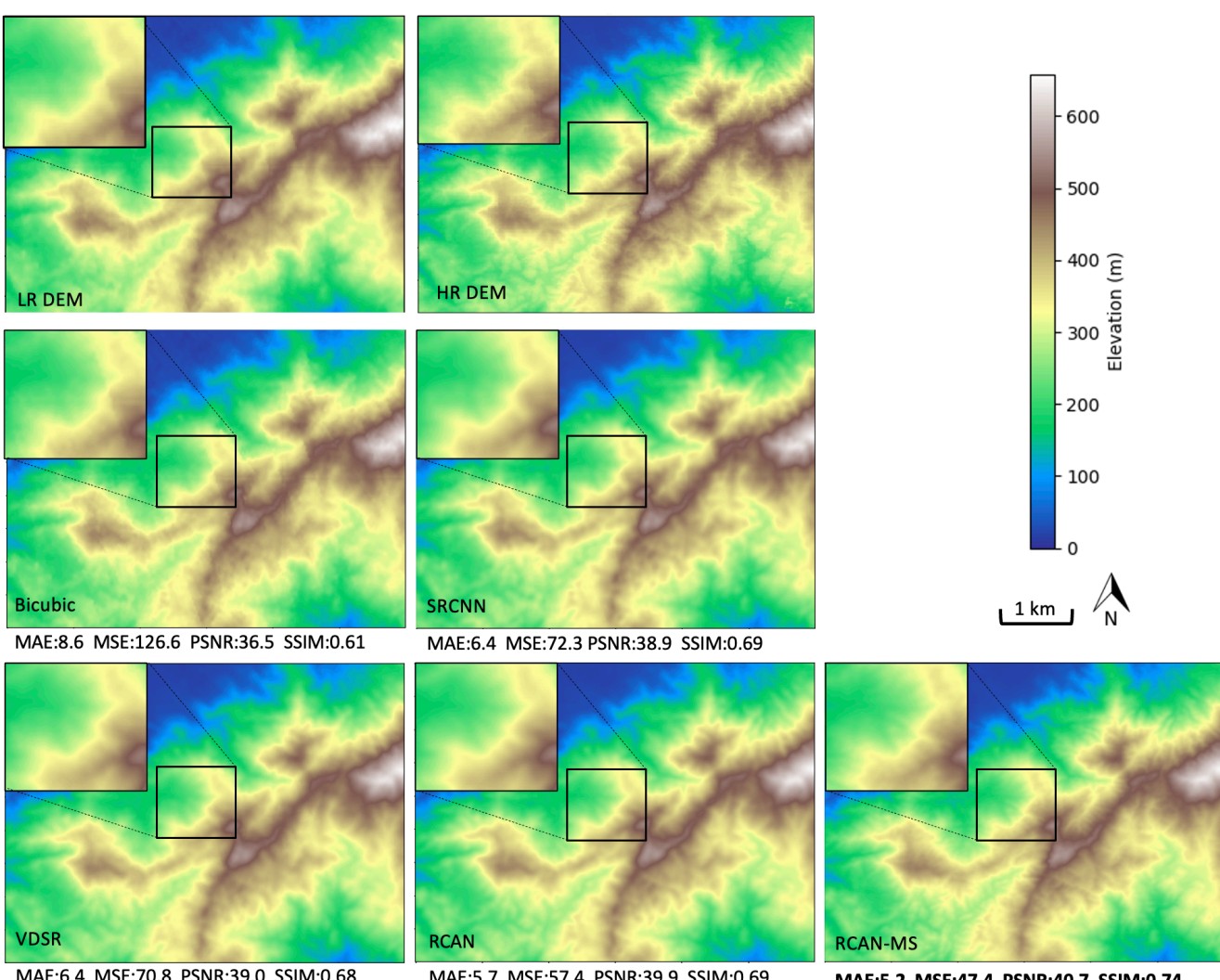

**Fig. 6 Comparison of DEM maps in the test set of Dataset 2. generated by the proposed method, RCAN-MS, other baseline methods and the original high-resolution DEM map in Dataset 2. The values of the evaluation metrics (i.e., MAE, MSR, PSNR, SSIM) for each method are presented.**

We note that the enlarged area of Dataset 2 is situated at a relatively higher elevation in the patch (Fig. 6). Despite the different geographical locations of the exemplary patches in the two datasets, the results of the DEM super-resolution test on Dataset 2 align with the results of Dataset 1. RCAN gained the second-best performance, and the proposed RCAN-MS still shows the best performance among the models tested, highlighting its effectiveness in reconstructing
fine-grained information and also capturing the complexity of terrain elevations.

In contrast to the exemplary patch from Dataset 2 (Fig. 6), the patch from Dataset 1 is characterized by a relatively flatter terrain (Fig. 5). Arguably, flatter areas could pose a greater challenge due to smaller variations in elevation, which are closer in magnitude to the vertical accuracy of the DEM, potentially increasing the likelihood of error. Given the superior overall performance of RCAN-MS in Dataset 1, this suggests its potential effectiveness in handling subtler
elevation changes. However, as we looked at two geographical regions only, the method performance in a wider range of terrain characteristics remains to be tested.

The conventional image interpolation method, bicubic, presents the worst performance in the exemplary patches from both Dataset 1 and Dataset 2 with pixelated DEM maps that lack fine-grained features. Regarding the deep-learning-based super-resolution methods in Dataset 1 (Fig. 5), the super-resolution DEM images generated by SRCNN
and VDSR exhibit over-smoothing effects, which lead to a loss of details. In contrast, the super-resolution DEM images produced by RCAN and RCAN-MS presented substantially less blurring effect than other baseline methods. Furthermore, by incorporating multispectral satellite images as part of the input, RCAN-MS generated a super-resolution DEM image with finer details in elevation difference compared with RCAN, which showed the best performance among all the baseline methods.

To assess the extent to which the improved DEM data can facilitate pluvial flood simulation, the exemplary patches of the two test sets were adopted as the input data in the pluvial flood simulations.

## 4.2 Pluvial flood simulation

Pluvial flood simulations were conducted using the super-resolution DEM data of the two exemplary patches presented in Fig. 5 and Fig. 6, including the DEM data produced by all the tested methods. As mentioned further above, the rainfall
scenario for pluvial flood simulation was set to be a 30-minute duration for a 100-year return period for both datasets, with intensities set according to the geographical location, respectively 42 mm/h for England in Dataset 1 and 190 mm/h for Hong Kong in Dataset 2.

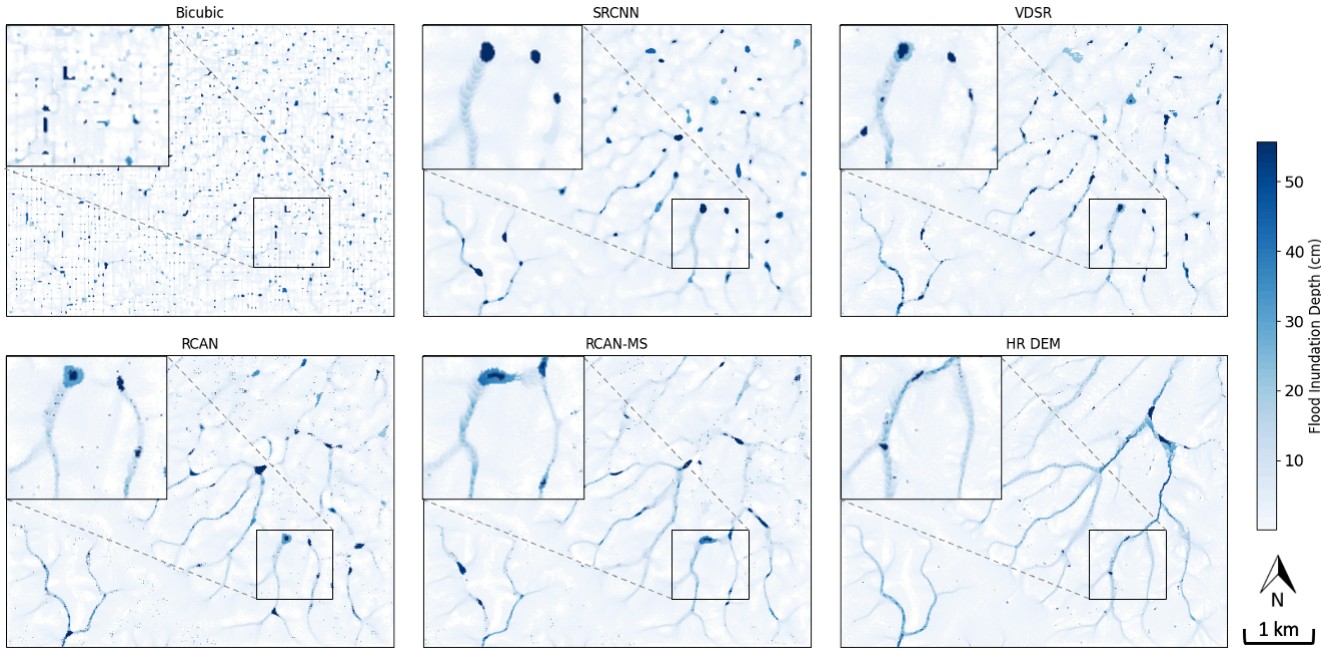

**Fig. 7 Maps of pluvial flood inundation depth simulated using super-resolution DEM data and compared with the flood inundation depth simulated using the original high-resolution DEM data in an exemplary patch of Dataset 1.**

Fig. 7 demonstrates the simulated pluvial floodwater inundation maps using super-resolution DEM maps of the exemplary patch of Dataset 1, which are illustrated in Fig. 5. Comparing the various flood inundation maps in Fig. 7, it

can be observed that the flood inundation map generated based on RCAN-MS replicates more similarly the floodwater distribution obtained using the high-resolution DEM, compared to the other considered super-resolution generating methods (i.e., Bicubic, SRCNN, VDSR, RCAN), thus reflecting the effects of smaller elevation errors of DEM in flood

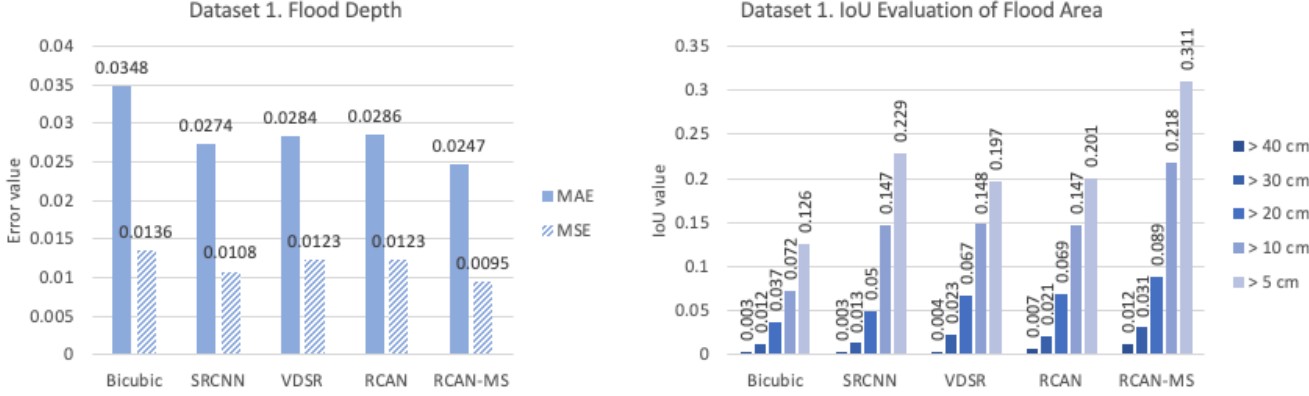

**Fig. 8 Performance evaluation of pluvial flood simulations based on super-resolution DEM data compared with the original high-resolution DEM data in the exemplary patch of Dataset 1. Left: MAE and MSE comparison of flood depth values; right: IoU evaluation of the spatial coverage of flood area delineated by different depth thresholds from 5cm to 40 cm.**

simulation. This can be particularly observed in the enlarged areas in Fig. 7, the inundated area predicted based on the RCAN-MS method exhibits greater consistency with the inundated area obtained using the high-resolution DEM, with
the distribution of floodwater appearing more contiguous, while other methods display more abrupt changes in water levels, which can be inferred from the scattered dark blue shading within the maps.

The performance in terms of floodwater depth and flood inundation area using super-resolution DEM data in the exemplary patch of Dataset 1 was also quantitatively evaluated and compared with the values obtained from simulations that used the reference high-resolution DEMs (Fig. 8). The errors of floodwater depth were evaluated using MAE and
MSE. The RCAN-MS method outperforms the other methods with the lowest MAE of 0.0247 m and the lowest MSE of 0.0095 m$^2$, scoring an approximately 30% improvement compared with conventional bicubic methods in both MAE and MSE. The accuracy of flood areas, defined using varying thresholds (i.e., 5 cm, 10 cm, 20 cm, 30 cm, 40 cm), was assessed using IoU, which measures the overlapping areas between the predicted flood areas defined using these thresholds. Here, the RCAN-MS method generally shows higher IoU values at different depth thresholds compared to
other methods, suggesting better precision in predicting the actual flood-affected area that will emerge from using the original DEM. Particularly for the flooded area with thresholds of 5 cm and 10 cm, the RCAN-MS method exhibits the highest IoU values with an improvement of 146% and 202% compared with bicubic interpolation.

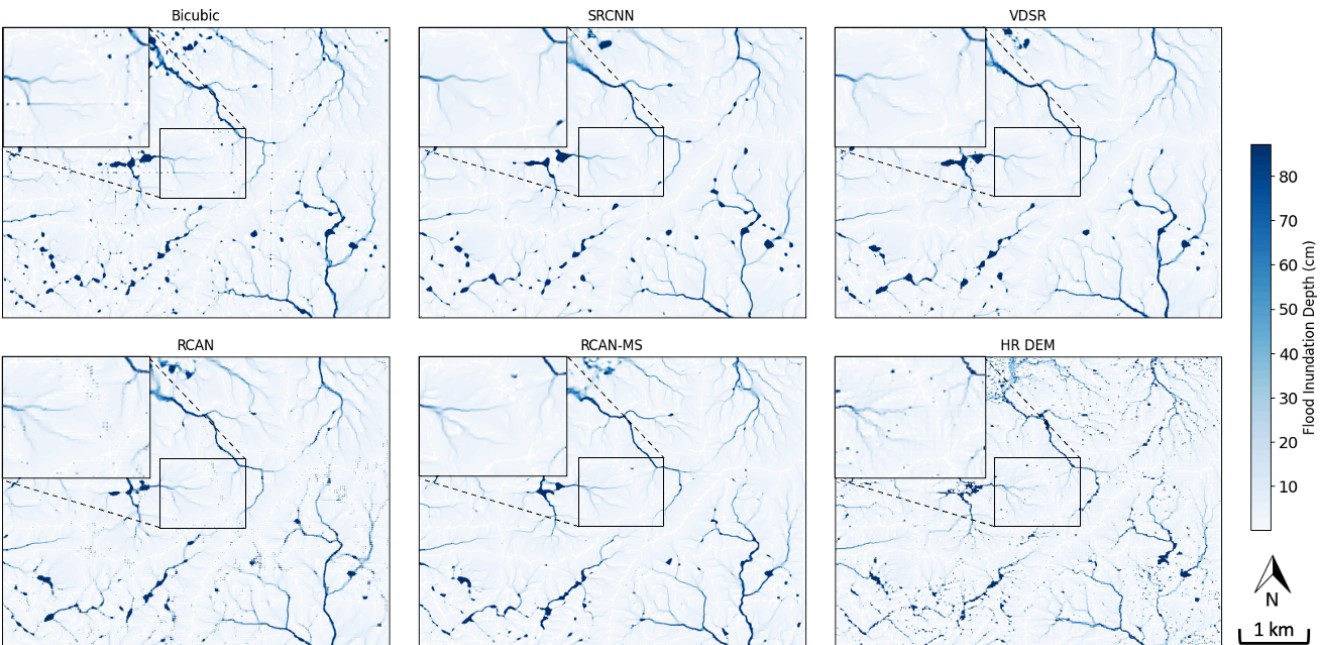

**Fig. 9 Maps of pluvial flood inundation depth simulated using super-resolution DEM data and compared with the flood inundation depth simulated using the original high-resolution DEM data in an exemplary patch of Dataset 2.**

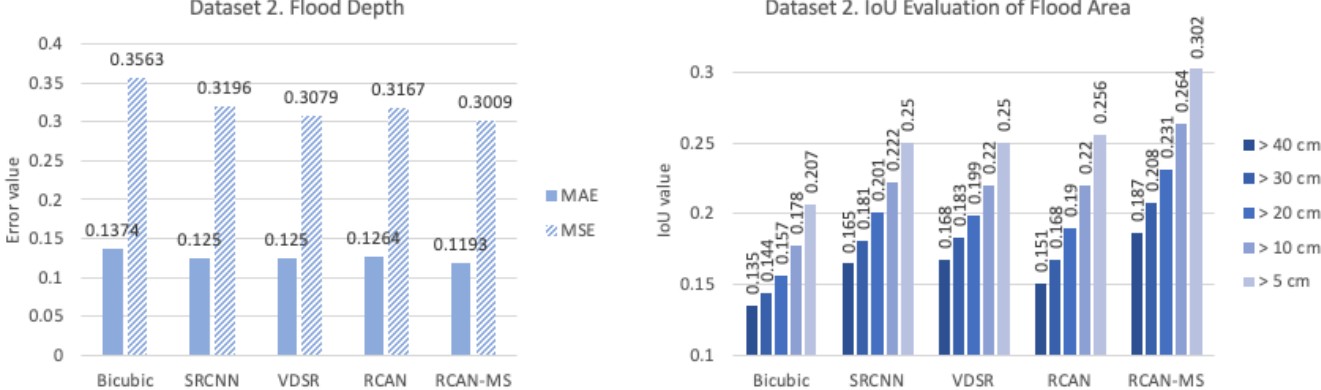

**Fig. 10 Performance evaluation of flood simulation maps produced based on super-resolution DEM data compared with the original high-resolution DEM data in the exemplary patch of Dataset 2. Left: MAE and MSE comparison of flood depth values; right: IoU evaluation of the spatial coverage of flood area delineated by different depth thresholds from 5cm to 40 cm.**

The comparisons of flood simulation results generated based on super-resolution DEM data from Dataset 2 are presented in Fig. 9 and Fig. 10. The visual inspection of flood inundation maps and quantitative evaluation suggest that results generally align with the performance obtained for Dataset 1. In particular, the flood inundation map generated based on RCAN-MS shows more fine-resolution details matching the floodwater distribution generated with the reference high-resolution DEM. Furthermore, the RCAN-MS-based flood inundation map yields the smallest MAE (0.1193 m) and MSE values (0.3009 m$^2$), indicating approximately 13% and 15% improvement in flood depth errors compared with the bicubic-based flood inundation map.

It can be observed in Fig. 8 and Fig. 10 that, although the proportional increase in IoU indicates that the proposed methods are correctly identifying more flood-prone areas compared to baseline methods, the IoU for high water depth thresholds is much lower than for lower water depth thresholds. This can be attributed to the significantly smaller spatial extent of deep floodwater areas. At higher thresholds, even small misalignments between the predicted and actual flood zones can result in a substantial reduction in IoU. While it becomes more challenging to simulate deep flood levels in their exact locations, flood simulation based on RCAN-MS still achieved the best performance in simulating deep floodwater areas compared to all baseline methods in both datasets.

**5 Discussion**

Among the investigated techniques to generate super-resolution DEMs, the here developed RCAN-MS provided superior performance compared to other baseline methods not only in terms of accuracy with respect to the high-resolution DEM, but also for the impact that its use has on flood simulation. Such superior performance is due to learning from multi-source inputs, particularly incorporating high-resolution multispectral satellite images, which enables it to achieve fine-

resolution details, while mitigating the pepper-and-salt noises in the super-resolution DEM generated by RCAN. Concurrently it avoids over-smoothing as instead occurs in the SRCNN and VDSR. Arguably, in RCAN-MS, the improvement effect of the multi-source input on DEM super-resolution is due to the input from multispectral information that contributes to a better understanding of how land cover features interact with different terrains, thus leading to more detailed and accurate terrain reconstructions (Chen et al., 2013). Specifically, the differentiation between vegetated areas and bare soil in multispectral data can increase the performance of the model in accurately predicting elevation changes and surface contours. The variety of spectral bands helps in distinguishing between features that may have similar elevation profiles but different spectral characteristics, such as the different inter-class variations between urban areas and rocky terrain.

Most importantly, the RCAN-MS method to build a high-resolution DEM substantially enhances the accuracy of flood simulations by producing DEM data with sufficient spatial resolution and improved terrain reconstructions. It is important to note that, in principle better performance in DEM super-resolution does not necessarily guarantee an improvement in flood simulation accuracy. This is evident in the experimental results that, although the backbone method RCAN achieved the second-best performance in the evaluation of DEM super-resolution tests in both datasets, RCAN fell short in pluvial flood simulation when compared to SRCNN and VDSR. This inferior performance can likely be attributed to the presence of pepper-and-salt noise within the flood inundation maps simulated from RCAN, where shallow-depth flooded pixels appear scattered. In contrast, SRCNN and VDSR, known for producing smoother ground surfaces, result in DEMs that lead to fewer instances of scattered floodwater pixels. Therefore, despite RCAN yielding fewer errors in DEM super-resolution compared to SRCNN and VDSR, the latter models achieve higher scores in simulated flood inundation maps due to the reduced occurrence of pepper-and-salt noise. This issue was substantially alleviated in the results of RCAN-MS due to the integration of multi-spectral satellite images. Such integration has proven to be effective in reducing noise and improving flood simulation accuracy. Incorporating additional data sources enables the model to better represent complex terrain features, which play crucial roles in flood simulation performance.

The terrain characteristics can influence the effectiveness of interpolation and super-resolution methods in flood simulation. Specifically, the improvement in flood simulation maps achieved by RCAN-MS is more evident in Dataset 1 than in Dataset 2. A key factor contributing to this discrepancy is the difference in terrain between the two datasets. As shown in Fig. 5 and Fig. 6, Dataset 1 features a relatively flat landscape, while Dataset 2 is characterised by hillier topography. In the flatter terrain of Dataset 1, floodwater tends to be distributed in a wider area, resulting in less distinct patterns and greater noise in the simulation results generated by baseline methods (e.g., bicubic interpolation). In contrast, the hilly terrain of Dataset 2 naturally promotes more concentrated flow accumulation, leading to visually coherent flood patterns across different methods, even using the DEM generated with bicubic interpolation. Therefore, the improvement

provided by the proposed super-resolution method tends to be more significant in regions with less pronounced topography.

While the present study shows the significantly better-performing nature of the RCAN-MS technique compared to other well-established methods, one should note that the study is also characterised by some limitations. First, for the two datasets tested in this study, as we intend to examine the performance of proposed models with a direct and efficient workflow, we did not apply data pre-processing techniques (e.g., noise reduction) on the high-resolution DEM data. There may still be room for improvement in flood simulations using super-resolution DEM data that have pre-processed

high-resolution DEM as training datasets. Also, it should be acknowledged that variations in acquisition dates of data from diverse sources can lead to minor inconsistencies in datasets. These temporal discrepancies, especially between multispectral satellite imagery, and between low-resolution and high-resolution images, may affect the efficacy of DEM super-resolution generation, potentially reducing its performance. In addition, one should observe that the model was trained and evaluated in two specific geographic areas. Thus, its straight transferability without minor adjustments (e.g.

fine-tuning of parameterisation) may not be guaranteed in other regions, particularly those with significantly different terrain characteristics. However, retraining or fine-tuning the model, which is generally possible, is expected to allow for effective implementation in many different regions.

      In future work, further tests could focus on investigating the impact of including additional inputs on model performance. This study takes advantage of multi-scale and multi-source input data for DEM super-resolution but only

incorporates 4-band multispectral satellite images as additional features. Other terrain-related features (e.g., slope, aspect) may potentially improve model performance and were not tested. Thus, future work can explore the impact of terrain-related features on enhancing model performance, as well as examine the performance of the proposed methods with different downscaling factors, where higher-resolution DEM data are available as training targets. Moreover, this study trained and evaluated models separately for two different regions characterised by different geographical and terrain

contexts. Future work, such as pretraining on diverse global DEM datasets and fine-tuning for specific local applications, could explore approaches to improve model generalizability, thus supporting broader applicability in regions with limited high-resolution DEM data. Furthermore, this study performed pluvial flood simulations using a cellular automata-based model forced with a rainfall scenario of a 1-in-100-year return period, further tests could assess the effects of super-resolution DEM under alternative rainfall scenarios and using additional flood simulation models to assess if DEM input

quality has some level of model dependency. While many flood models share the same fundamental equations to solve for flow processes (Guo et al., 2021), such extended analyses would likely broaden the scope of the study and enhance the generalisability of the results.

## 6 Conclusion

This study addresses the critical challenge of accurate flood simulation in regions where high-resolution DEM data is
unavailable or of limited extent. We developed and implemented a deep learning-based DEM super-resolution method,
incorporating multi-source input data, including low-resolution DEM and high-resolution multispectral imagery. The
experiment suggests that the enhanced multi-source DEM super-resolution method, RCAN-MS, significantly improves
the accuracy of the DEM for pluvial flood simulations, particularly in terms of floodwater depth and inundation area
predictions. The integration of Sentinel-2A multispectral data with the 30m SRTM DEM allows for the reconstruction
of 10m DEM data with higher fidelity compared to conventional methods. The improved performance of RCAN-MS in
flood simulation, compared to its backbone method RCAN, underscores the value of incorporating multispectral images
as they enhance terrain representation and reduce noise in the super-resolution DEM, thus leading to more accurate flood
simulation results.

By leveraging publicly available global datasets, this approach offers a promising solution for regions with
limited availability to high-resolution topographic data, enabling not only more precise flood simulations, but also the
potential to generate large-scale high-resolution DEMs from existing publicly available coarse DEMs and, thus, opens
new possibilities for resilience development and resource allocation, potentially not only contributing to flood risk
reduction, but also to broader applications in simulating other natural hazards where accurate terrain representation is
essential.

**Data and Code availability.** Except for the data from TanDEM-X, which requires a proposal submission and approval for
data acquisition, all other data and codes are openly accessible here: https://zenodo.org/records/15212783. The software for
cellular automata-based pluvial flood simulation can be downloaded from the CADDIES website
(https://www.exeter.ac.uk/research/centres/cws/resources/caddies/).

**Author contributions.** All authors contributed to the idea and scope of the paper. Yue Zhu contributed to conceptualisation,
data curation, formal analysis, methodology, data analysis, software and script, validation, and original draft writing. Paolo

Burlando, Puay Yok Tan, and Simone Fatichi contributed to supervision, review and editing, and funding acquisition. Christian Geiß contributed to data acquisition and supervision.

**Conflict of Interest.** No conflict of interest amongst the authors.

## Acknowledgements

This research was funded in part by the Future Cities Lab Global programme. Future Cities Lab Global is supported and funded by the National Research Foundation, Prime Minister's Office, Singapore under its Campus for Research Excellence and Technological Enterprise (CREATE) programme and ETH Zurich (ETHZ), with additional contributions from the National University of Singapore (NUS), Nanyang Technological University (NTU), and the Singapore University of Technology and Design (SUTD).

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
