# Peer review of "Improving Pluvial Flood Simulations with Multi-source DEM Super-Resolution"

_Natural Hazards and Earth System Sciences, 2024_

## Referee Comment (RC2)

[referee-annotated manuscript omitted]

---

## Author Comment (AC1)

**The paper introduces a deep-learning method that combines low resolution DEM and multi-spectral images to obtain a high-resolution DEM that is ultimately used for running a pluvial flood simulation. The authors also compare this approach with other DL and not methods showing its improved efficacy.**

**The manuscript is well written, clear, concise, and informative. As such I recommend publication with just few minor details that might further improve the quality of the paper.**

*Thank you very much for your comments, we provide a detailed point-by-point response to each comment as below.*

**Minor comments:**

1. **In the results/discussion section, I would emphasize that the difference between the RCAN and the RCAN-MS is mainly in the inputs used (if I followed everything correctly), thus further proving your point that the extra information coming from multi-spectral images is beneficial, since so far it seemed "just" a difference in method as you have with VDSR, for example.**

*Thank you for this suggestion. We agree with this and will emphasise in the manuscript that the primary distinction between RCAN and RCAN-MS lies in the inputs used, as the tailored input layers to processing multi-sourced inputs. This added information supports the advantage of using multi-spectral images and strengthens the argument that they contribute to improved model performance. We can revise the discussion section to clarify this as follows:*

*(Line 138) "This study adopts a multi-source method for DEM super-resolution, utilizing the RCAN as the backbone structure. The proposed method, referred to as RCAN-Multispectral (RCAN-MS), incorporates a tailored multi-source and multi-scale input module, which is the key distinction from the original RCAN."*

*(Line 147) "The tailored multi-source input module is integrated into the model structure before the first layer of the RCAN backbone structure (Fig. 1)."*

*(Line 400) "The improved performance of RCAN-MS in flood simulation, compared to its backbone method RCAN, underscores the value of incorporating multispectral data. The additional information provided by the multispectral images enhances terrain representation and reduces noise in the super-resolution DEM, thus leads to more accurate flood simulation results."*

2. **Could you explain why do the results in terms of flood simulations look more consistent in Dataset II rather than in Dataset I, at least visually? For example, in Figure 6, all interpolation methods seem to produce some sort of accumulation ponds in correspondence of the bifurcations of the rivers and the bicubic approximation results in a noisy pattern. However, that does not seem the case for Figure 8 with Dataset II. Do you have any clue why?**

*Thank you for raising this point. A potential explanation for the difference in flood simulation results between the two datasets may stem from the terrain characteristics of the study areas. As shown in Figures 4 and 5, the test area in Dataset 1 is relatively flat, while the second test area in Dataset 2 has a hillier terrain. In Dataset 1, the flatter landscape leads to a more diffuse distribution of floodwater, which can result in less distinct patterns and variability in the simulation results. In contrast, the hilly*

*terrain of Dataset 2, even with bicubic interpolation, naturally facilitates more concentrated floodwater accumulation in certain areas, resulting in relatively more consistent simulation outcomes across different methods.*

*We can include this discussion in the manuscript as follows (line 360):*

*"In addition, the terrain characteristics can influence the effectiveness of interpolation and super-resolution methods in flood simulation. Specifically, the improvement in flood simulation maps achieved by RCAN-MS is more pronounced in Dataset 1 than in Dataset 2. A key factor contributing to this discrepancy is the difference in terrain between the two datasets. As shown in Fig. 5 and Fig. 6, Dataset 1 features a relatively flat landscape, while Dataset 2 is characterized by hillier topography. In the flatter terrain of Dataset 1, floodwater tends to be more diffusely distributed, resulting in less distinct patterns and greater noise in the simulation results generated by baseline methods (e.g., bicubic interpolation). In contrast, the hilly terrain of Dataset 2 naturally promotes more concentrated water accumulation in specific areas, leading to more visually coherent flood patterns across different methods, even with bicubic interpolation. Therefore, the improvement provided by the proposed super-resolution method tends to be more significant in flatter regions, where its effects are more pronounced."*

3. **I think you could also comment further on why is the IoU very low (despite the proportional increase) for high thresholds of water depths.**

*Thank you for your question. The low IoU for high water depth thresholds, despite the proportional increase, can likely be attributed to the much smaller extent of deep floodwater areas. At higher thresholds, the areas of flooding become more concentrated in specific regions with much smaller spatial coverage, which may not align well with the predicted flood areas. In this case, at higher depth thresholds, even small misalignments between the predicted and actual flood zones can result in a significant decrease in IoU. While the proportional increase suggests that the model is correctly identifying more flood-prone areas as the water depth threshold rises, the precision and spatial accuracy required to match the predicted and actual flood extents become more challenging.*

*We can make the corresponding revision in the manuscript as follows:*

*(line 325) "It can be observed in Fig. 8 and Fig. 10 that, although the proportional increase in IoU indicates that the proposed methods are correctly identifying more flood-prone areas compared to baseline methods, the IoU for high water depth thresholds is much lower than for lower water depth thresholds. This can be attributed to the significantly smaller spatial extent of deep floodwater areas. At higher depth thresholds, even small misalignments between the predicted and actual flood zones can result in a substantial reduction in IoU. While it becomes more challenging to simulate flood extents at higher depth thresholds, flood simulation based on RCAN-MS still achieved the best performance in simulating deep floodwater areas compared to all baseline methods in both datasets."*

4. **In terms of metrics you could also consider adding a different metric such as the critical success index (CSI), which has been used in several flood studies.**

*Thank you for this suggestion. We incorporated Intersection over Union (IoU) as one of the metrics in our analysis. The formulas for IoU and Critical Success Index (CSI) are mathematically identical in the context of this study. Both metrics measure the overlap between the predicted and actual positive areas (True Positives, TP) relative to the total number of areas covered by both predicted positives (TP + FP) and actual positives (TP + FN), which can be expressed as:*

$$CSI = IoU = \frac{TP}{TP + FP + FN}$$

*We believe this provides an adequate measure of overlap and performance in our flood simulation results.*

5. **While most figures are of high quality, I think Figure 7 and 9 can be better, despite already being informative. Consider changing their style.**

*To improve Fig 7 and 9 (which are now Fig 8 and 10 in the revised manuscript), we changed the figure style to bar charts as follows:*

[Figure]

**Fig. 8 Performance evaluation of pluvial flood simulations based on super-resolution DEM data compared with the original high-resolution DEM data in the exemplary patch of Dataset I. Left: MAE and MSE comparison of flood depth values; right: IoU evaluation of the spatial coverage of flood area delineated by different depth thresholds from 5cm to 40 cm.**

[Figure]

**Fig. 10 Performance evaluation of flood simulation maps produced based on super-resolution DEM data compared with the original high-resolution DEM data in the exemplary patch of Dataset II. Left: MAE and MSE comparison of flood depth values; right: IoU evaluation of the spatial coverage of flood area delineated by different depth thresholds from 5cm to 40 cm.**

---

## Author Comment (AC2)

**RC3**: 'Comment on nhess-2024-207'

This paper presents a new deep learning method to downscale coarse, satellite-derived terrain data to 10m resolution by exploiting higher resolution multispectral image data. The results of the method are validated for two case areas, both through direct comparison against high resolution terrain data, and by comparing pluvial flood simulations with varying terrain inputs. Several benchmark downscaling methods are included in the comparison.

I think this is a very good paper. I very much appreciate the investigation of the effects of downscaling methods on the final application, i.e., pluvial flood simulation, and I think it is well placed within the scope of the journal. I have only some very minor comments that are mentioned below and don't require further review. I suggest accepting the paper.

Comments:

1. Language - please perform a proofread, there are several typos distributed throughout the paper

*Thank you for your comments. We performed a careful proofreading and corrected typos in this manuscript.*

2. Units - please include units in the results figures e.g. Fig. 7 and 9. Similarly, the scores in Table 2 require units. I believe that the test in Hongkong does not have an average error of 8m, but how should we interpret an MSE of 66???

*Thank you for pointing this out. The metrics in Table 2 require proper units. PSNR and SSIM are unitless. MAE is measured in meters (m), MSE is measured in square meters ($m^2$), and reflects the mean of squared elevation differences, making it more sensitive to outliers. The MSE of 66.6251 aligns with the average MAE of 5.8181 when considering that MSE is more sensitive to outliers. As such, MSE provides a measure of error variability, where large errors have more influence. This highlights the importance of interpreting both MAE and RMSE for a complete understanding of model performance. We can includ these units in the revised table to clarify the interpretations.*

**Table 1. Evaluation results of all the tested methods on two test sets with different geographical locations**

|  | *Test set of Dataset 1.* | | | | *Test set of Dataset 2.* | | | |
|---|---|---|---|---|---|---|---|---|
|  | **MAE (m)** | **MSE (m²)** | **PNSR** | **SSIM** | **MAE (m)** | **MSE (m²)** | **PNSR** | **SSIM** |
| bicubic | 3.0078 | 19.0206 | 33.4055 | 0.4621 | 9.2924 | 163.0170 | 35.4505 | 0.6091 |
| SRCNN | 2.7665 | 15.5027 | 34.2901 | 0.5776 | 6.8153 | 94.1950 | 37.8500 | 0.6794 |
| VDSR | 2.6530 | 13.4866 | 34.8653 | 0.5737 | 6.6412 | 88.7638 | 38.1110 | 0.6811 |
| RCAN | 2.5967 | 12.9453 | 35.0460 | 0.5975 | 6.4150 | 83.5288 | 38.3950 | 0.6838 |
| RCAN-MS | **2.1952** | **8.7102** | **36.7605** | **0.6205** | **5.8181** | **66.6251** | **39.3543** | **0.7411** |

**3. Figure 1 - please include resolutions in the figure. The entire residual in residual block operates in 30m resolution. In addition, the upscaling module is not described. I suppose this is another 2D convolution. Does it receive a skip connection with high resolution as input?**

*Thank you for this comment.*

*In terms of the resolution in Figure 1, we have included resolution information for the low-resolution DEM data, high-resolution multi-spectral satellite images, and the super-resolution DEM output. We argue that, since the data passing through convolutional layers are tensors that may not have explicit physical units, it is more appropriate to represent the height and width of the spatial dimensions as spatial resolution in the figure. Therefore, we can add the spatial resolution (height, width) in this figure, as shown below:*

[Figure]

**Fig. 1 The structure of the proposed DEM Super-resolution model, MS-RCAN. Low-resolution DEM data and 4-band multispectral satellite images are adopted as the input to reconstruct high-resolution DEM data.**

*Regarding the upscaling module, we can add a description regarding the upscaling layer as follows:*

*(line 155) "After that, the concatenated multi-source input is passed through the RCAN backbone structure, which consists of RIR blocks and includes a 2D convolutional layer at the end of the model structure to upscale the data flow to the size of the high-resolution DEM map."*

*We did not add a skip connection between the input and the high-resolution output, this is because there is a long skip connection at the end of the input module and before the upscaling layer at the end of the model structure, which are just a few layers to reach the final output.*

---

## Author Comment (AC3)

We sincerely appreciate the time and effort the reviewers invested in evaluating our manuscript and providing insightful comments. We have carefully considered each comment. To facilitate the review process, we are including a detailed point-by-point response to the comments, indicating how each issue has been addressed. In the below point-by-point response, our responses are in *italic*, and the text in the revised manuscript is marked with quotation marks. Thank you for considering our revised submission.

**Attached are additional comments. Thank you for the nice paper, it was a pleasure to read.**

*We will make the following response to the comments and revisions to the manuscript:*

*(line 15) we will modify the abstract with a clarification about the research context: "Accurate flood simulation remains a significant challenge in many flood-prone regions, particularly in developing countries and urban areas, where the availability of high-resolution topographic data is especially limited." and about the quantification of performance metrics: "We evaluated the performance of the super-resolution DEM in flood simulations. Compared to conventional methods (e.g., bicubic interpolation), the simulation results demonstrated that our approach significantly improved the accuracy of flood simulations, with a reduction in the mean absolute error of floodwater depth from 0.137 to 0.119 (-13.1%) and an increase in the IoU for inundation area predictions from 0.207 to 0.302 (+45.9%)."*

*(line 25) We will tone down the introduction as: "The occurrence of severe floods has been on the rise, partly influenced by climate change, which contributes to more frequent extreme rainfall events (Tabari, 2020)."*

*(line 30) We will add the citation: "open datasets of DEM data with global coverage are predominantly available at raster resolutions coarser than 30 x 30 meters (Marsh et al., 2023)"*

*(line 36) We can rephrase the introduction to existing methods for improving the spatial resolution of DEM to differentiate with other similar statements: "Methods to enhance the spatial resolution of DEM data have been widely adopted across various geospatial applications to improve risk estimation. These advancements have significantly enhanced the accuracy and reliability of natural hazard mapping, including flood prediction (Löwe & Arnbjerg-Nielsen, 2020; Tan et al., 2024), landslide modelling (Brock et al., 2020), volcanic flow assessment (Deng et al., 2019), and snow avalanche forecasting (Miller et al., 2022)."*

*(line 46) We can rephrase the introduction to data fusion-based approaches for improving DEM spatial resolution for better clarity: "During the fusion process, tools such as elevation error maps or weight maps are commonly used to assign importance to each DEM source, ensuring that higher-quality data has a greater influence on the final output. However, these methods often introduce inaccuracies by altering elevation values and failing to address edge effects (Okolie & Smit, 2022), such as abrupt transitions or mismatches between overlapping DEM datasets."*

*(line 52) As suggested, we will remove the acronym SISR for Single Image Super-Resolution due to its infrequent use in the manuscript. We did not adopt the acronym SR for super-resolution, as it consists of only two letters and could be confused with spatial resolution, which is frequently referenced throughout the manuscript.*

*(line 50) we consolidated the sentence introducing deep-learning DEM super-resolution methods: "The implementations of deep learning-based super-resolution methods have been shown to substantially improve the performance of remote sensing applications (Ling & Foody, 2019; Shang et al., 2022; Xie et al., 2022) and promote the utilisation of data that was previously underutilised due to limited spatial resolution (Zhu et al., 2021), including the applications of enhancing low-resolution DEM data (Demiray et al., 2021a, 2021b; Jiang et al., 2023; Kubade et al., 2020; Z. Li et al., 2023; Yue et al., 2015; Zhou et al., 2023, 2021, 2021)."*

*(line 60) We will revise the literature review to provide a richer discussion of related studies, elaborating on their methods and performance. Additionally, we explained how our approach improves upon these previous works: "For instance, Demiray et al. (2021) utilized Generative Adversarial Networks (GANs) to upscale low-resolution DEMs (50ft) to high-resolution DEMs (3ft), although this study demonstrated the potential of adversarial training in spatial resolution enhancement, GANs are known for unstable in training, facing challenges such as mode collapse and vanishing gradients (Jabbar et al., 2021). Zhou et al. (2021) introduced a double-filter deep residual neural network, leveraging residual learning to improve feature extraction and enhance the accuracy of reconstructed DEMs. More recently, Li et al. (2023) proposed a transformer-based deep learning network for upscaling DEM across multiple upsampling factors (e.g., ×2, ×4), showcasing the effectiveness of attention mechanisms in capturing long-range dependencies and spatial relationships. Building on the advances of these existing methods, we refine a DEM super-resolution method by employing a computationally efficient architecture with attention mechanisms to achieve accuracy and robustness."*

*(line 65) We will rewrite the section "1.2. Multi-source deep learning for remote sensing applications" to remove redundant or unnecessary sentences and added related citations. The modified paragraphs are as follows:*

*"The benefits of integrating multi-source inputs in remote sensing applications have been increasingly recognised, as the combination of complementary data sources enhances the robustness and reliability of model performance (J. Li et al., 2022). For instance, Shen et al. (2019) developed a deep learning-based model for drought monitoring, which employed multi-source remote sensing data as input, including DEM data, and meteorological and soil data. Lu et al. (2022) proposed a deep learning framework taking Google Earth imagery and point of interest heatmap as input data for urban functional zone extraction. Blöschl et al. (2024) integrated additional bathymetric information into the DEM to enhance national-scale flood hazard mapping.*

*With respect to the input for DEM super-resolution, it can be argued that, solely replying to a single source of low-resolution (LR) DEM input can be an ill-posed task, as high-resolution details can hardly be accurately reconstructed without additional reference information (Yue et al., 2016). Studies have been made to include additional features generated from low-resolution DEM data. For instance, Zhang et al. (2023) calculated terrain gradient maps based on DEM data to guide the optimisation process of a Convolutional Neural Network (CNN)-based DEM super-resolution. Zhou et al. (2023) proposed a terrain feature-based CNN for DEM super-resolution, which extracts slope and aspect from low-resolution DEM data and deploys them as additional features for model inputs and loss function.*

*Besides generating additional features based on low-resolution DEM, efforts have also been made to fuse different data sources to offer fine-granular details related to terrain features to bring performance gains. One example following this direction is found in Argudo et al. (2018), who examined the feasibility of combining natural colour aerial images together with low-resolution DEM data as input to train a CNN for producing high-resolution DEM, suggesting improved performance compared with interpolation-based methods. Tan et al. (2024) introduced a deep learning-based DEM upscaling network that uses high-resolution optical images to predict elevation differences, and then fuses these predictions with the original DEM data through additional convolutional layers. It should*

*be noted that these studies mainly employed natural colour images for feature fusion. In contrast, multispectral images can provide further features from non-visible wavelengths, such as near-infrared, allowing for more detailed and specialised analysis. This is supported by Chen et al. (2013), showcasing the effects of utilising multispectral bands of satellite images on improving the performance of an interpolation-based DEM densification method. More recently, a few attempts have explored the effects of integrating low-resolution DEM with remote sensing imagery for DEM super-resolution. Gao & Yue (2024) used the red band of Sentinel-2 images to provide auxiliary high-frequency information for DEM super-resolution training. Paul & Gupta (2024) incorporated 3-band satellite images with low-resolution DEM to develop a GAN-based DEM super-resolution model."*

*(line 100) We will rephrase the second main contribution of the study as: "...(ii) the use of publicly open datasets ensures the generalizability of the method, especially for DEM-related applications in data-scarce regions;"*

*(line 110) Thank you for the comments on revising the overall significance of the study from "offering an exemplary pathway to address the issue of lacking high-resolution DEM for reliable risk assessments in the context of land use planning and disaster management" to focus on emphasising "improve flood simulation". However, given that we intend to keep the section on pluvial flood simulation evaluation in the main manuscript, we would like to argue that the section on quantifying the improvements in pluvial flood simulation also indicates the potential of improving broader applications across various domains that rely on high-resolution DEMs for reliable spatial analysis. Therefore, we think the original statement reflects the broader significance of the study, showcasing its value beyond flood simulations and positioning it as a methodological advancement applicable to multiple disciplines.*

*(line 110) We thank you for the comment about adding sentences guiding the reader into the method section. Accordingly, we will add a guiding sentence as follows: "To improve the spatial resolution of DEM data for enhancing flood simulations, we proposed a deep learning-based DEM super-resolution method. This method employs the Residual Channel Attention Network (RCAN) (Y.Zhang et al., 2018) as the backbone structure and incorporates a tailored multi-source input block to leverage multi-sourced input data, contributing to improved performance in reconstructing high-resolution DEM data."*

*(line 120) As suggested, we will modify the last sentence of this paragraph to clarify our modifications on the backbone structure as follows: "However, since RCAN is developed for image super-resolution tasks on single natural colour images, we tailored the structure of its input module to handle inputs from different data sources."*

*(line 125) As suggested, we will modify section 2.2 "Multi-source and multi-scale input data fusion" to avoid redundancy in method descriptions as follows: "This study adopts a multi-source method for DEM super-resolution, utilizing the RCAN as the backbone structure. The proposed method, referred to as RCAN-Multispectral (RCAN-MS), incorporates a tailored multi-source and multi-scale input module, which is the key distinction from the original RCAN. This input module enables the integration of high-resolution multispectral satellite images with low-resolution DEM data, leveraging the complementary information from both sources to reconstruct high-resolution DEMs with enhanced accuracy. Multispectral satellite images contain information captured across various spectral bands, including both visible light and invisible bands, which offer a wealth of information about surface materials, vegetation coverage, water bodies, and other landscape features (Carrão et al., 2008), making them ideal for compensating for the coarse information in low-resolution DEMs. By combining high-resolution multispectral imagery with low-resolution elevation data, deep learning models can access a more comprehensive feature set, facilitating the reconstruction of detailed topographic information."*

*(line 135) Thank you for the suggestion, we will revise Figure 1 to emphasise the location of the multi-source input module in the model structure, which now corresponds to the text: "The tailored multi-source input module is integrated into the model structure before the first layer of the RCAN backbone structure (**Error! Reference source not found.**)."*

[Figure]

**Fig. 1 The structure of the proposed DEM super-resolution model, MS-RCAN. Low-resolution DEM data and four-band multispectral satellite images are fused using a tailored multi-source and multi-scale input module to facilitate the reconstruction of high-resolution DEM data.**

**Comment: need to expand this section so it is more clear what your contribution/improvements are vs. the original RCAN framework.**

*(line 145) Thank you for the comment on clarifying that the main difference between the proposed method and RCAN is the multi-source and multi-scale input module. This is clarified in the earlier sections as follows: "The proposed method, referred to as RCAN-Multispectral (RCAN-MS), incorporates a tailored multi-source and multi-scale input module, which is the key distinction from the original RCAN."*

*(Table 1) Thank you for the comments on Table 1, we added citations to the corresponding dataset in the table, the revised version is as reported here below:*

**Table 1. Information on the DEM data and multispectral satellite images in two datasets for the tests of DEM super-resolution models**

|  |  | Dataset 1. England | Dataset 2. Shenzhen & Hong Kong |
|---|---|---|---|
| 10m DEM | Collection source | LIDAR Composite DTM 2019, published by UK Environment Agency (2023) | TanDEM-X, provided by German Aerospace Centre (DLR)) |
|  | Spatial resolution | Resampled from 2m to 10m resolution using a bilinear interpolation | Resampled from 12m to 10m resolution using a bilinear interpolation |
|  | Acquisition date | 2019-09-01 | 2016-01-13 |
| 30m DEM | Collection source | Shuttle Radar Topography Mission (SRTM), accessed from USGS EarthExplorer | Shuttle Radar Topography Mission (SRTM) , accessed from USGS EarthExplorer |
|  | Spatial resolution | 1 arc-second ($\sim$ 30m) resolution | 1 arc-second ($\sim$ 30m) resolution |
|  | Acquisition date | 2014-09-23 | 2014-09-23 |
| 10m Multispectral Images | Collection source | Sentinel-2A | Sentinel-2A |
|  | Spatial resolution | 10m resolution | 10m resolution |
|  | Bands | Band 2 – Blue, Band 3 – Green, Band 4 – Red, Band 8 - Near-infrared | Band 2 – Blue, Band 3 – Green, Band 4 – Red, Band 8 - Near-infrared |
|  | Acquisition date | 2022-11-25 / 2023-01-21/ 2023-02-13 | 2023-12-25 |

*NASA JPL (2013). NASA Shuttle Radar Topography Mission Global 1 arc second [Data set]. NASA EOSDIS Land Processes Distributed Active Archive Center. Retrieved from https://doi.org/10.5067/MEaSUREs/SRTM/SRTMGL1.003*

*UK Environment Agency. (2023). LIDAR Composite DTM 2019 – 10m. Retrieved from https://www.data.gov.uk/dataset/8311f42d-bddd-4cd4-98a3-e543de5be4cb/lidar-composite-dtm-2019-10m*

*(line 160) As suggested, we will add more information on the data sources as follows:*

*"All the data in these two datasets are collected from publicly open sources, including SRTM, TanDEM-X, and Sentinel-2, which have been widely adopted for remote sensing applications in urban environments (Wu, et al., 2019; Geiß et al., 2015; C. Li, et al., 2021). SRTM utilized dual radar antennas to collect interferometric radar data, which was then processed into digital topographic data with a resolution of 1 arc-second (Farr et al., 2007). TanDEM-X mission uses a single-pass interferometric synthetic aperture radar (InSAR) system to produce 12 m resolution global digital surface models. The Sentinel-2 satellites carry the Multi-Spectral Instrument (MSI), which captures imagery in 13 spectral bands, with the blue, green, red, and near-infrared bands having a 10m spatial resolution (Spoto et al., 2012)."*

*Thank you for the suggestions for improving Fig.2. We changed the colours of DEM maps to ensure a colour-blind-friendly appearance, and added north arrows, using the same colour scale for similar maps. We did not add a legend for multi-spectral images as they are presented in false colour mode with the composition of three bands. Additionally, we added a world map and marked the location of the two datasets. The modified figure is as follows:*

[Figure]

**Fig. 2 Overview of the two datasets for DEM Super-resolution. (a) the training, validation, and test sets of Dataset 1. (b) the training, validation, and test sets of Dataset 2 (see Table 1 for data source).**

*Comment:* **what did you do with the edge/leftover pixels?**

*(line 170) We subsample patches in the areas with valid pixel values, therefore no leftover pixels were involved in the training process.*

**Comment: missing a lot of description that are needed to make your method reproduceable (e.g., hyperparameters, data augmentation, framework (pytorch?), architecture/layer sizes)**

*(line 175) we have responded in the above main comment section, in which we revised the manuscript to provide details regarding training settings and hyperparameters.*

*Regarding **the comment on the necessity of the pluvial flood simulation section**, as addressed in the main comment section, a key contribution of this study is to quantify the impact of the super-resolution DEM, generated using the proposed deep-learning method, on hazard simulations. This quantification offers two main benefits: (i) Since the evaluation metric of super-resolution DEM may not necessarily reflect their effectiveness in geospatial applications, evaluating how DEMs generated through different methods perform in end applications, particularly in flood hazard modelling, can provide more comprehensive performance evaluation. (ii) For practitioners in the field of hazard applications, this section offers insight into whether the proposed deep-learning approach provides a cost-efficient solution for enhancing the accuracy and applicability of flood simulations. We added the above arguments in the manuscript as follows:*

*(line 176) "The first stage was centred on assessing the performance of DEM Super-resolution methods in enhancing the resolution of the original DEM data, whereas the second stage was to quantify the effects of adopting the super-resolution DEM on enhancing pluvial flood simulations. This quantification offers two main benefits: (i) provides a more comprehensive performance evaluation on how DEMs generated through different methods perform in end applications; (ii) examines whether the proposed deep-learning approach provides a cost-efficient solution for improving flood simulations."*

*For the **comments in section 3.2 Experiment setup**, we provide the answers as follows (line 185):*

- *Regarding the training setup of baseline models, we adopted the same training setup as the proposed method;*
- *The models in the two case studies were trained separately, but using the consistent training setup described in the manuscript;*
- *Regarding the selection of batch size and learning rate, we conducted experiments with various configurations of batch sizes and learning rates. The tests indicated that the chosen configuration achieved the best performance. Additionally, we employed an adaptive learning rate scheduler, which reduced the learning rate by a factor of 0.8 when the validation loss did not decrease for 50 epochs.*

**Comment: I'd like to see a plot of MAE vs. epoch for all relevant models on both datasets. I think this will make the methods clearer.**

[Figure]

**Fig. 3 Changes in the MAE values of all the tested models as training epochs increase for Dataset 1 (left) and Dataset 2 (right).**

*As requested, we can add a figure to present MAE vs. epoch for all relevant models on both datasets as below:*

**Comment on section 4.1 DEM Super-resolution: there are too many acronyms and numbers here for me to follow... unless you are pointing out something special.. just leave the info in the table. Or talk about 'percent change' if you want to be more quantitative.**

*We will revise this paragraph by emphasising just the most important numerical values and adding percentage changes. The revised paragraph is as follows:*

*(line 215) "For Dataset 1, the RCAN-MS method demonstrates a marked improvement over the Bicubic method, reducing the MAE from 3.0 to 2.2 (-26.7%), and the MSE from 19.0 to 8.7 (-54.2%). This enhancement is also reflected in the values of PSNR (+9.9%) and SSIM (+34.8%), suggesting a substantially improved fit to the target high-resolution DEM. Similarly, Dataset 2 results reveal that RCAN-MS significantly outperforms the bicubic interpolation method, with the MAE sharply decreasing from 9.9 to 5.9 (-40.4%), and the MSE from 186.0 to 67.6 (-63.7%). The RCAN method, serving as the backbone method for RCAN-MS, shows better results than the other deep learning-based methods such as SRCNN and VDSR across both datasets, underscoring the superior performance of the RCAN-based architecture in the task of DEM super-solution. Specifically, for Dataset 1, RCAN posts an MAE of 2.60 and an MSE of 12.9, which are better than those for SRCNN and VDSR. In Dataset 2, RCAN achieves an MAE of 6.4 and an MSE of 83.5, further confirming its robustness. The performance superiority of the proposed RCAN-MS method is evident across all metrics in both datasets, demonstrating its enhanced capability in generating high-fidelity super-resolution DEM data. This is exemplified by the significant reductions in MAE and MSE and the corresponding increase in PNSR and SSIM values, signifying its substantial improvements over the baseline methods."*

**Comment on Table 2: show units**

*Thank you for this comment, we will add units for MAE and MSE in the table. However, PNSR is unitless ratio, and SSIM a similarity measure ranging from 0 to 1. The revised table can be as below:*

**Table 2. Evaluation results of all the tested methods on two test sets with different geographical locations**

|  | Test set of Dataset 1. | | | | Test set of Dataset 2. | | | |
|---|---|---|---|---|---|---|---|---|
|  | MAE (m) | MSE (m²) | PNSR | SSIM | MAE (m) | MSE (m²) | PNSR | SSIM |
| bicubic | 3.0078 | 19.0206 | 33.4055 | 0.4621 | 9.2924 | 163.0170 | 35.4505 | 0.6091 |
| SRCNN | 2.7665 | 15.5027 | 34.2901 | 0.5776 | 6.8153 | 94.1950 | 37.8500 | 0.6794 |
| VDSR | 2.6530 | 13.4866 | 34.8653 | 0.5737 | 6.6412 | 88.7638 | 38.1110 | 0.6811 |
| RCAN | 2.5967 | 12.9453 | 35.0460 | 0.5975 | 6.4150 | 83.5288 | 38.3950 | 0.6838 |
| RCAN-MS | **2.1952** | **8.7102** | **36.7605** | **0.6205** | **5.8181** | **66.6251** | **39.3543** | **0.7411** |

**Comment regarding experimental results represented in Fig 5 and Fig 6:**

- **use the same color pallete as your previous figure**

*We would like to argue that, since the previous figure, Figure 2, adopts a color-blind friendly palette, the contrast of values difference is less visually detectable. Unlike Figure 2, which presents an overview of the two datasets, Figures 5 and 6 compare the super-resolution DEMs generated by different models, which contain subtle differences. To make the differences more distinctive, a color palette with multi-hued transitions is more effective. Therefore, we use the 'terrain' color palette in Matplotlib for these two figures.*

- **include the SRTM source image (I assume this will be identical to 'bicubic'.. so just ammend the axis title). and an optical image**

*We will amend Figures 5 and 6 to include SRTM source images.*

- **this figure (Figure 5) has a lot of redundant information as**

[Figure]

**Fig. 4 Comparison of DEM maps in the test set of Dataset 1. generated by the proposed method, RCAN-MS, other baseline methods and the original high-resolution DEM map in Dataset 2.**

[Figure]

**Fig. 5 Comparison of DEM maps in the test set of Dataset 1. generated by the proposed method, RCAN-MS, other baseline methods and the original high-resolution DEM map in Dataset 1.**

Thank you for pointing this out. We removed the scale bar on each subplot, and only kept one for all of them on the right side of the figure.

- **for the zoom/call-out, use a different colorscale so we can see what is going on**

*Thank you for this suggestion. The primary objective of the zoom-in patch is to enlarge details within the tested area while maintaining consistency with the multi-hued color palette used for the main plots across both datasets. Our current approach ensures that variations remain detectable without introducing potential misinterpretations from an altered color scale. While applying a different color scale to the zoom-in plots might add additional contrast, it could also disrupt consistency and comparability. Given these considerations, we believe our current visualization effectively conveys the necessary details.*

- **oh.. I see you are reporting the metrics separately for the patch.. is this needed?**

*Table 2 provides the overall performance metrics for the entire test set, which may not capture differences in specific subareas. Therefore, to facilitate a meaningful comparison between these patches, we also report their corresponding metrics. This approach ensures that the patches are quantitatively comparable.*

- **I think this whole paragarph could be replaced with "test patch performs similarly"... unless there is some performance difference we should be aware of.... this also seems like it can be removed (paragraph starting with "The enlarged area of Dataset II....")**

*Thank you for this suggestion, we will remove some numerical descriptions that present consistent performance with the overall evaluation on the test set, whereas we preferred to preserve the description related to the pluvial flood simulation. The revised paragraph is as below:*

*(line 240) "Fig. 4 and **Error! Reference source not found.** present the two selected patches from the test sets of Datasets 1 and 2 for visual assessment of the performance of the super-resolution DEM maps, in which a subarea of exemplary patches is additionally enlarged for further visual comparison of details. The corresponding reference low-resolution DEM and high-resolution DEM map are also presented for comparison. The values of the evaluation metrics (i.e., MAE, MSR, PSNR, SSIM) are also summarised in the two figures. The ranking of the performance of all the tested methods is aligned with the overall evaluation of the test sets reported in **Error! Reference source not found.**, suggesting t hat RCAN offers a larger magnitude of enhancement than SRCNN and VDSR, and RCAN-MS stands out among all the tested methods, recording the lowest MAE and MSE values. These two exemplary patches of the test sets are employed for pluvial flood simulation in the following section.*

*The enlarged area of Dataset 2 is situated at a relatively higher elevation in the patch (**Error! R eference source not found.**). Despite the different geographical locations of the exemplary patches in the two datasets, the results of the DEM super-resolution test on Dataset 2 align with the results of Dataset 1. RCAN gained the second-best performance, and the proposed RCAN-MS also presents in the case of Dataset 2 the best performance among the models tested, highlighting its effectiveness in reconstructing fine-grained information and also capturing the complexity of terrain elevations."*

- **I don't think you're 1 patch comparison is enough evidence to claim that your model performs better in flat terrain. include some discussion of this limitation here.**

*Thank you for pointing this out. We would like to clarify that the proposed model not only achieved the best performance on this single patch but also demonstrated superior overall performance on Dataset I, which generally features relatively flatter terrain compared to Dataset II. However, we agree that this does not definitively confirm its generalizability in flat terrains. Therefore, we will tone down the statement and added a discussion of potential limitations.*

*(line 255) "In contrast to the exemplary patch from Dataset 2 (Fig. 6), the patch from Dataset 1 is characterized by a relatively flatter terrain (Fig. 5). Arguably, flatter areas could pose a greater challenge due to smaller variations in elevation, which are closer in magnitude to the vertical accuracy of the DEM, potentially increasing the likelihood of error. Given the superior overall performance of RCAN-MS in Dataset 1, this suggests its potential effectiveness in handling subtler elevation changes. However, the datasets only represent terrains from two geographical regions, which do not encompass the full diversity of terrain characteristics."*

- **Please add an evaluation of cross-validating the two datasets (i.e., use the Hong Kong model weights to make predictions in England). This will better communicate the methods ability for transferability.**

*Thank you for your suggestion. We acknowledge that an extended cross-validation between the two datasets may provide additional insights into model transferability. However, differences in data distribution, sensor resolution, and regional terrain features between the two datasets may introduce confounding factors that require further adaptations. In this sense, applying a model trained on one specific source domain directly to another without any adaptations (e.g., parameter tuning) could*

*significantly impact performance, particularly given the distinct terrain characteristics of the two regions.*

*We understand that extended cross-validation between models trained in different datasets would be a valuable approach for studies prioritising transferability. For instance, a truly generalisable model would require training on diverse datasets that minimize bias rather than relying on a single source domain. However, our primary focus in this study is to assess model performance within different datasets under consistent training conditions, ensuring applicability to various terrain characteristics. The experimental results on both datasets have met these objectives.*

**Comment on Section 4.2 Pluvial flood simulation: I suggest moving all of this to the supplement**

*As clarified in previous responses, we consider the pluvial flood simulation section as essential because it quantifies the impact of super-resolution DEMs on hazard modeling, complementing the evaluation metrics of DEM super-resolution and underlying the practical implications in hazard modelling. This section provides indeed practitioners with insights into the cost-effectiveness of the proposed deep-learning approach for improving flood simulation accuracy. Therefore, we prefer to retain this section in the main text.*

**Comment on Fig 9: need to make it more clear that these are popout boxes; use a different color for zero**

*Thank you for this comment, we will amend the figures using white color for zero and made them clearer as pop-out boxes. The revised figures are as below:*

[Figure]

**Fig. 6 Maps of pluvial flood inundation depth were simulated using super-resolution DEM data and compared with the original high-resolution DEM data in an exemplary patch of Dataset 1.**

[Figure]

**Fig. 7 Maps of pluvial flood inundation depth were simulated using super-resolution DEM data and compared with the original high-resolution DEM data in an exemplary patch of Dataset 2.**

**Comment on Section 5. Discussion:**

- **"pepper-and-salt noise effects" need to talk about this in the introduction**

*Thank you for this comment. We would like to clarify that the "pepper-and-salt noise effects" mentioned here refer to issues observed in the super-resolution DEM data generated by RCAN, one of the baseline models. This is not a common issue for this model, nor is it a characteristic of the original data. Therefore, we believe it is more appropriate to discuss this in the discussion section rather than in the introduction. To avoid any misunderstanding, we have revised the sentence for better clarity. The revised version is as follows:*

*(line 305) "Such superior performance is due to learning from multi-source inputs, particularly the incorporation of high-resolution multispectral satellite images, enabling it to achieve fine-resolution details, while mitigating the pepper-and-salt noises in the super-resolution DEM generated by RCAN but avoiding over-smoothing in SRCNN and VDSR."*

- **"in RCAN-MS, the improvement effect of the multi-source input on DEM super-resolution is likely due to the additional feature extracted from multispectral information" didn't you show this? This is why you need to provide more information on your baseline models.**

*We wish to clarify that "the additional features" here refer to the multi-spectral satellite images that are employed as part of the input for RCAN-MS, this has been explained in the previous sections. To avoid unambiguity, we will revise this description for better clarity as follows:*

*(line 306) "in RCAN-MS, the improvement effect of the multi-source input on DEM super-resolution is likely due to the input from multispectral information, which provides additional features to facilitate the estimation of the reflectance of varying land cover types."*

- **"in principle better performance in DEM super-resolution does not necessarily guarantee an improvement in flood simulation accuracy" This is an interesting point, and a good**

**argument for including some sort of hydrologic metrics in your evaluation (as you did with the pluvial flooding). However, I still think this is a minor point... esp as it only supports the conclusion of your traditional metrics.**

*Thank you for pointing this out. As clarified in earlier responses to this concern, besides supporting traditional metrics, an important contribution of having the section on pluvial flood simulation is to quantify the extent to which super-resolution DEM generated by the proposed model can improve flood simulation, compared with other baseline models. This not only can highlight the effectiveness of the method in hazard simulation applications, but also provide important reference for practitioners to consider and evaluate the cost-effective of this approach in similar applications.*

- **"only incorporated 4-band multispectral satellite images as additional features, other terrain-related features (e.g., slope, aspect) that may bring further improvement to model performance were not tested in this study" this isn't really a limitation.. more about future work**

*Thank you for this comment. We will revise this paragraph to merge into the section of discussion on future work. The revised content is as follows:*

*(line 335) "In future work, further tests could focus on investigating the impact of including additional features on model performance. This study takes advantage of multi-scale and multi-source input data for DEM super-resolution but only incorporates 4-band multispectral satellite images as additional features. Other terrain-related features (e.g., slope, aspect) that may improve model performance were not tested. Thus, future work can explore the impact of terrain-related features on enhancing model performance, as well as examine the performance of the proposed methods with different downscaling factors, where higher-resolution DEM data is available as training targets."*

- **"In future work…" this is better placed in the conclusion section.**

*We believe this discussion of future studies is better suited for the discussion section rather than the conclusion. As the discussion of future studies here is more connected with the above-described limitations, putting it in the discussion section allows for a critical reflection on the limitations and highlights areas for further exploration. Meanwhile, we tend to have the conclusion section focus on summarizing the key findings and main takeaways. Introducing future work in the conclusion may detract from the focus on the study results.*

**Comment: Need to have a very good reason to not share your code. Esp. considering all the authors seem to come from publicly funded institutions. I can not complete the review without seeing the code.**

*As responded in the previous section of the main comments, we agree with openly sharing the code and data, except for the high-resolution DEM data for Dataset 2. This information can be added in the manuscript as follows:*

*"Except for the data from TanDEM-X, which requires a proposal submission and approval for data acquisition, all other data and codes are openly accessible here: https://zenodo.org/records/14868516"*

---

## Author Response (AR1)

Dr. Yue Zhu Future Cities Laboratory Singapore-ETH Centre 06-01, 1 Create Wy Singapore 138602 e-mail: yue.zhu@sec.ethz.ch

Editorial Board Natural Hazards and Earth System Sciences

Re.: Submission of Revised Manuscript

Dear Sir/Madam,

I am writing to submit the revised version of our manuscript titled "Improving Pluvial Flood Simulations with Multi-source DEM Super-Resolution" for consideration in Natural Hazards and Earth System Sciences.

We sincerely appreciate the time and effort the reviewers invested in evaluating our manuscript and providing insightful comments. We have carefully considered each comment and have made corresponding revisions to the manuscript.

To facilitate the review process, we are including a detailed point-by-point response to the comments, indicating how each issue has been addressed. In the below point-by-point response, our responses are in *italic*, and the text in the revised manuscript is marked with quotation marks. Changes made to the manuscript are highlighted in the track-changes file.

We believe that these revisions have further enhanced our manuscript and hope that it will be considered favourably for publication. Thank you for considering our revised submission. Please do not hesitate to contact me if further information is needed.

Sincerely, Yue Zhu

**Referee #1**

The paper introduces a deep-learning method that combines low resolution DEM and multi-spectral images to obtain a high-resolution DEM that is ultimately used for running a pluvial flood simulation. The authors also compare this approach with other DL and not methods showing its improved efficacy.

The manuscript is well written, clear, concise, and informative. As such I recommend publication with just few minor details that might further improve the quality of the paper.

Thank you very much for your comments, we have made corresponding revisions to the manuscript, and provide a detailed point-by-point response to each comment as below.

**Minor comments:**

1. In the results/discussion section, I would emphasize that the difference between the RCAN and the RCAN-MS is mainly in the inputs used (if I followed everything correctly), thus further proving your point that the extra information coming from multi-spectral images is beneficial, since so far it seemed "just" a difference in method as you have with VDSR, for example.

Thank you for this suggestion. We agree with this and have emphasised in the manuscript that the primary distinction between RCAN and RCAN-MS lies in the inputs used, as the tailored input layers to processing multi-sourced inputs. This added information supports the advantage of using multi-spectral images and strengthens the argument that they contribute to improved model performance. We revised the discussion section to clarify this as follows:

(Line 138) "The proposed method, referred to as RCAN-Multispectral (RCAN-MS), incorporates a tailored multi-source and multi-scale input module, which is the key distinction from the original RCAN."

(Line 147) "The tailored multi-source input module is integrated into the model structure before the first layer of the RCAN backbone structure (Fig. 1)."

(Line 395) "The improved performance of RCAN-MS in flood simulation, compared to its backbone method RCAN, underscores the value of incorporating multispectral images as they enhance terrain representation and reduce noise in the super-resolution DEM, thus leading to more accurate flood simulation results."

2. Could you explain why do the results in terms of flood simulations look more consistent in Dataset II rather than in Dataset I, at least visually? For example, in Figure 6, all interpolation methods seem to produce some sort of accumulation ponds in correspondence of the bifurcations of the rivers and the bicubic approximation results in a noisy pattern. However, that does not seem the case for Figure 8 with Dataset II. Do you have any clue why?

Thank you for raising this point. A potential explanation for the difference in flood simulation results between the two datasets may stem from the terrain characteristics of the study areas. As shown in Figures 4 and 5, the test area in Dataset 1 is relatively flat, while the second test area in Dataset 2 has a hillier terrain. In Dataset 1, the flatter landscape leads to a more diffuse distribution of floodwater, which can result in less distinct patterns and variability in the simulation results. In contrast, the hilly terrain of Dataset 2, even with bicubic interpolation, naturally facilitates more concentrated floodwater accumulation in certain areas, resulting in relatively more consistent simulation outcomes across different methods.

We include this discussion in the manuscript as follows:

(line 350) "The terrain characteristics can influence the effectiveness of interpolation and superresolution methods in flood simulation. Specifically, the improvement in flood simulation maps achieved by RCAN-MS is more evident in Dataset 1 than in Dataset 2. A key factor contributing to this discrepancy is the difference in terrain between the two datasets. As shown in Fig.5 and Fig. 6, Dataset 1 features a relatively flat landscape, while Dataset 2 is characterized by hillier topography. In the flatter terrain of Dataset 1, floodwater tends to be distributed in a wider area, resulting in less distinct patterns and greater noise in the simulation results generated by baseline methods (e.g., bicubic interpolation). In contrast, the hilly terrain of Dataset 2 naturally promotes more concentrated flow accumulation, leading to visually coherent flood patterns across different methods, even using the DEM generated with bicubic interpolation. Therefore, the improvement provided by the proposed super-resolution method tends to be more significant in regions with less pronounced topography."

**3. I think you could also comment further on why is the IoU very low (despite the proportional increase) for high thresholds of water depths.**

Thank you for your question. The low IoU for high water depth thresholds, despite the proportional increase, can likely be attributed to the much smaller extent of deep floodwater areas. At higher thresholds, the areas of flooding become more concentrated in specific regions with much smaller spatial coverage, which may not align well with the predicted flood areas. In this case, at higher depth thresholds, even small misalignments between the predicted and actual flood zones can result in a significant decrease in IoU. While the proportional increase suggests that the model is correctly identifying more flood-prone areas as the water depth threshold rises, the precision and spatial accuracy required to match the predicted and actual flood extents become more challenging.

We made the corresponding revision in the manuscript as follows:

(line 316) "It can be observed in Fig. 8 and Fig. 10 that, although the proportional increase in IoU indicates that the proposed methods are correctly identifying more flood-prone areas compared to baseline methods, the IoU for high water depth thresholds is much lower than for lower water depth thresholds. This can be attributed to the significantly smaller spatial extent of deep floodwater areas. At higher thresholds, even small misalignments between the predicted and actual flood zones can result in a substantial reduction in IoU. While it becomes more challenging to simulate deep flood levels in their exact locations, flood simulation based on RCAN-MS still achieved the best performance in simulating deep floodwater areas compared to all baseline methods in both datasets."

**4. In terms of metrics you could also consider adding a different metric such as the critical success index (CSI), which has been used in several flood studies.**

Thank you for this suggestion. We incorporated Intersection over Union (IoU) as one of the metrics in our analysis. The formulas for IoU and Critical Success Index (CSI) are mathematically identical in the context of this study. Both metrics measure the overlap between the predicted and actual positive areas (True Positives, TP) relative to the total number of areas covered by both predicted positives (TP + FP) and actual positives (TP + FN), which can be expressed as:

$$CSI = IoU = \frac{TP}{TP + FP + FN}$$

We believe this provides an adequate measure of overlap and performance in our flood simulation results.

**5. While most figures are of high quality, I think Figure 7 and 9 can be better, despite already being informative. Consider changing their style.**

To improve Fig 7 and 9 (which are now Fig 8 and 10 in the revised manuscript), we changed the figure style to bar charts as follows:

Fig. 8 Performance evaluation of pluvial flood simulations based on super-resolution DEM data compared with the original highresolution DEM data in the exemplary patch of Dataset I. Left: MAE and MSE comparison of flood depth values; right: IoU evaluation of the spatial coverage of flood area delineated by different depth thresholds from 5cm to 40 cm.

Fig. 10 Performance evaluation of flood simulation maps produced based on super-resolution DEM data compared with the original high-resolution DEM data in the exemplary patch of Dataset II. Left: MAE and MSE comparison of flood depth values; right: IoU evaluation of the spatial coverage of flood area delineated by different depth thresholds from 5cm to 40 cm.

**Referee #2**

The authors extend the practice of super-resolution (SR) for DEM by including multi-spectral image inputs. By enhancing the resolution of SRTM using this Sentinel-2A data, they demonstrate the performance improvement provided by their method for 2 case studies. The study is extended by comparing the performance of the resulting DEMs in a pluvial flood simulation.

 From what I can tell, this is a well-done study, advances SR-DEM, and should be published. However, I'm not sure the work fits within NHESS... rather than journals more focused on ML or hydrodynamic modelling, like those where the references studies are published. i.e., there is only one NHESS reference in the bib (and this ref is intro fluff not related to the work). But maybe this is more of an editorial decision.

Thank you for raising this point, we tend to believe that, as NHESS is an interdisciplinary journal publishing research with topics related to various aspects of natural hazards, studies on investigating input data quality to facilitate enhanced hazard mapping fit the journal's scope. We also found some other published NHESS papers in this field. For instance, Blöschl et al. (2024) investigated hyper-resolution flood hazard mapping, which involves enhanced DEM data for improved flood simulation. Miller et al. (2022) tested the impact of different spatial resolutions of DEM data on snow avalanche modelling. Löwe & Arnbjerg-Nielsen (2020) explored the effect of data resolution on urban pluvial flood risk assessment.

We added the related studies as references in the manuscript as follows:

(line 40) "Methods to enhance the spatial resolution of DEM data have been widely adopted across geospatial applications to improve risk estimates. These advancements have significantly enhanced the accuracy and reliability of natural hazard mapping, including flood prediction (Löwe & Arnbjerg-Nielsen, 2020; Tan et al., 2024), landslide modelling (Brock et al., 2020), volcanic flow assessment (Deng et al., 2019), and snow avalanche forecasting (Miller et al., 2022)."

(line 75) "In general remote sensing applications, the benefits of integrating multi-source inputs have been increasingly recognised, as the combination of complementary data sources enhances the robustness and reliability of model performance (J. Li et al., 2022). ... Blöschl et al. (2024) integrated additional riverbed geometry information into the DEM to enhance national-scale flood hazard mapping."

- 2. Further, the authors could consider the following suggestions:
  - I appreciate the use of pluvial flood simulations to generate an additional performance metric for the SR, however I think this work should be de-emphasized and moved to the supplement... leaving the manuscript more focused on the SR architecture and experiment (which should be better described). i.e., while the pluvial sections take up roughly half the current manuscript, this work does not really influence the conclusion or abstract. You could instead focus on how traditional raster metrics (e.g., MAE) are inadequate for flood simulations... this would be an interesting paper... but a different paper.

Thank you for this comment. We would like to argue that the section on pluvial flood simulation evaluation should remain in the main manuscript. This is because the study aims to improve pluvial flood simulation by enhancing DEM data through super-resolution techniques, addressing the critical

issue of the lack of publicly available high-resolution DEMs for flood mapping. Examining the effect of super-resolution DEM data on flood simulation is essential to the objectives of this study. Notably, quantifying the extent to which the proposed method improves flood simulation provides valuable insights for other researchers and practitioners considering whether to adopt this approach in their studies and work. Furthermore, the experimental results in the flood simulation section demonstrate that better performance in DEM super-resolution, as measured by traditional metrics, does not necessarily lead to improved performance in flood simulation. Therefore, to validate the effectiveness of the proposed super-resolution approach in enhancing flood simulation, it is crucial to include the flood simulation analysis as an integral part of this study.

**• Code and data should be made public. e.g., there is no way to properly review such a manuscript without access to the code and data.**

Thank you for raising this point. We agree to openly share the code and data of this study, except for the high-resolution data for Dataset 2, which is from TanDEM-X. This is because TanDEM-X data requires proposal submission and approval for data acquisitions. This information is added in the manuscript as follows:

"Except for the data from TanDEM-X, which requires a proposal submission and approval for data acquisition, all other data and codes are openly accessible here: https://zenodo.org/records/14868516"

• The comparison against 4 baseline methods is a great idea and communicates the benefits of the proposed method well. However, these baseline methods (like the proposed method) need to be described adequately for reproducibility. i.e., the training and hyperparameters used. The authors should take care to provide as 'fair' a comparison as possible.

Thank you for this suggestion, we revised the manuscript to provide more details on training strategies and hyperparameters as follows:

(line 205) "The training of DEM Super-resolution models was established and trained with PyTorch on two NVIDIA GeForce RTX 4090 GPUs on high-performance computing (HPC) clusters. All baseline models were implemented using the default parameter settings for hidden layers as specified in their original papers. The input and output layer configurations were adapted to suit the task of DEM superresolution. The baseline methods used as benchmarks utilized single-band input and output layers, except for RCAN-MS, which was configured with five input bands (i.e., single-band DEM and four-band multispectral image). All the test methods adopted the same training strategy, they were all trained with a batch size of 8 and a learning rate of 1×10-4. With an adaptive learning rate scheduler, the learning rate decreases to a fraction of 0.8 when the validation loss stops decreasing for 50 epochs. The optimizer adopted for all the methods is Adam with default momentum parameters. The loss function is the Mean Absolute Error (MAE). Regarding the stopping criteria for mode performance evaluation, all the models were trained for 200 epochs with the data in the training set, after which the epoch yielding the smallest MAE values on the validation set was selected for further performance elevation on the test sets."

• The transferability of the method to different regions should be better explored and discussed (does the model need to be retrained for each region?)

Thank you for highlighting this important aspect. Our model is trained and evaluated separately on data from specific study areas, and while the results demonstrate its effectiveness in this context, its transferability to other geographic areas has not been thoroughly tested, and it is expected that certain retraining or parameter-tuning would be required to achieve a good performance in other regions. Therefore, we addressed this point as a limitation in the manuscript as follows:

(line 370) "In addition, one should observe that the model was trained and evaluated in two specific geographic areas. Thus, its straight transferability without minor adjustments (e.g. fine tuning of parameterisation) may not be guaranteed in other regions, particularly those with significantly different terrain characteristics. However, retraining or fine-tuning of the model, which is generally possible, is expected to allow for effective implementation in many different regions."

**3. Attached are additional comments. Thank you for the nice paper, it was a pleasure to read.**

Thank you for the detailed comments in the attachments, which are very helpful in improving the quality of this study. We made the following response to the comments and revisions to the manuscript:

(line 15) we modified the abstract with a clarification about the research context: "Accurate flood simulation remains a significant challenge in many flood-prone regions, particularly in developing countries and urban areas, where the availability of high-resolution topographic data is especially limited." and about the quantification of performance metrics: "We evaluated the performance of the super-resolution DEM in flood simulations. Compared to conventional methods (e.g., bicubic interpolation), the simulation results demonstrated that our approach significantly improved the accuracy of flood simulations, with a reduction in the mean absolute error of floodwater depth of about 13.1% and an increase in the IoU for inundation area predictions of about 46%."

(line 28) We have toned down the introduction as: "The occurrence of severe urban floods has been on the rise, partly influenced by climate change, which contributes to more frequent extreme rainfall events (Tabari, 2020)."

(line 33) We added the citation: "At present, open datasets of DEM data with global coverage are predominantly available at raster resolutions coarser than (or equal) 30m (Marsh et al., 2023)"

(line 40) We rephrased the introduction to existing methods for improving the spatial resolution of DEM to differentiate with other similar statements: "Methods to enhance the spatial resolution of DEM data have been widely adopted across various geospatial applications to improve risk estimation. These advancements have significantly enhanced the accuracy and reliability of natural hazard mapping, including flood prediction (Löwe & Arnbjerg-Nielsen, 2020; Tan et al., 2024), landslide modelling (Brock et al., 2020), volcanic flow assessment (Deng et al., 2019), and snow avalanche forecasting (Miller et al., 2022)."

(line 50) We rephrased the introduction to data fusion-based approaches for improving DEM spatial resolution for better clarity: "During the fusion process, tools such as elevation error maps or weight maps are commonly used to assign importance to each DEM source, ensuring that higher-quality data has a greater influence on the final output. However, these methods often introduce inaccuracies by altering elevation values and failing to address edge effects (Okolie & Smit, 2022), such as abrupt transitions or mismatches between overlapping DEM datasets."

(line 55) As suggested, we removed the acronym SISR for Single Image Super-Resolution due to its infrequent use in the manuscript. We did not adopt the acronym SR for super-resolution, as it consists

of only two letters and could be confused with spatial resolution, which is frequently referenced throughout the manuscript.

(line 60) we consolidated the sentence introducing deep-learning DEM super-resolution methods: "The implementations of deep learning-based super-resolution methods have been shown to substantially improve the performance of remote sensing applications (Ling & Foody, 2019; Shang et al., 2022; Xie et al., 2022) and promote the utilisation of data that was previously underutilised due to limited spatial resolution (Zhu et al., 2021), including the applications of enhancing low-resolution DEM data (Demiray et al., 2021a, 2021b; Jiang et al., 2023; Kubade et al., 2020; Z. Li et al., 2023; Yue et al., 2015; Zhou et al., 2023, 2021, 2021)."

(line 65) We revised the literature review to provide a richer discussion of related studies, elaborating on their methods and performance. Additionally, we explained how our approach improves upon these previous works: "For instance, Demiray et al. (2021) utilized Generative Adversarial Networks (GANs) to upscale low-resolution DEMs (50ft) to high-resolution DEMs (3ft), although this study demonstrated the potential of adversarial training in spatial resolution enhancement, GANs are known for unstable in training, facing challenges such as mode collapse and vanishing gradients (Jabbar et al., 2021). Zhou et al. (2021) introduced a double-filter deep residual neural network, leveraging residual learning to improve feature extraction and enhance the accuracy of reconstructed DEMs. More recently, Li et al. (2023) proposed a transformer-based deep learning network for upscaling DEM across multiple upsampling factors (e.g., ×2, ×4), showcasing the effectiveness of attention mechanisms in capturing long-range dependencies and spatial relationships. Building on the advances of these existing methods, we refine a DEM super-resolution method by employing a computationally efficient architecture with attention mechanisms to achieve accuracy and robustness."

(line 75) We rewrote the section "1.2. Multi-source deep learning for remote sensing applications" to remove redundant or unnecessary sentences and added related citations. The modified paragraphs are as follows:

"In general remote sensing applications, the benefits of integrating multi-source inputs have been increasingly recognised, as the combination of complementary data sources enhances the robustness and reliability of model performance (J. Li et al., 2022). For instance, Shen et al. (2019) developed a deep learning-based model for drought monitoring, which employed multi-source remote sensing data as input, including DEM data, and meteorological and soil data. Lu et al. (2022) proposed a deep learning framework taking Google Earth imagery and point of interest heatmap as input data for urban functional zone extraction. Blöschl et al. (2024) integrated riverbed geometry information into the DEM to enhance national-scale flood hazard mapping.

With respect to the input for DEM super-resolution, it can be argued that, solely relying on a single source of low-resolution (LR) DEM input can be an ill-posed task, as high-resolution details can hardly be accurately reconstructed without additional reference information (Yue et al., 2016). Studies have been made to include additional features generated from low-resolution DEM data. For instance, Zhang et al. (2023) calculated terrain gradient maps based on DEM data to guide the optimisation process of a Convolutional Neural Network (CNN)-based DEM super-resolution. Zhou et al. (2023) proposed a terrain feature-based CNN for DEM super-resolution, which extracts slope and aspect from low-resolution DEM data and deploys them as additional features for model inputs and loss function.

Besides generating additional features based on low-resolution DEM, efforts have also been made to fuse different data sources to offer fine-granular details related to terrain features, which can improve performance. One example following this direction is found in Argudo et al. (2018), who examined the feasibility of combining natural colour aerial images together with low-resolution DEM data as input

to train a CNN for producing high-resolution DEM, suggesting improved performance compared with interpolation-based methods. Tan et al. (2024) introduced a deep learning-based DEM upscaling network that uses high-resolution optical images to predict elevation differences, and then fuses these predictions with the original DEM data through additional convolutional layers. It should be noted that these studies mainly employed natural colour images for feature fusion. In contrast, multispectral images can provide further features from non-visible wavelengths, such as near-infrared, allowing for more detailed and specialised analysis. This is supported by Chen et al. (2013), showcasing the effects of utilising multispectral bands of satellite images on improving the performance of an interpolation-based DEM densification method. More recently, a few attempts have explored the effects of integrating low-resolution DEM with remote sensing imagery for DEM super-resolution. Gao & Yue (2024) used the red band of Sentinel-2 images to provide auxiliary high-frequency information for DEM super-resolution training. Paul & Gupta (2024) incorporated 3-band satellite images with low-resolution DEM to develop a GAN-based DEM super-resolution model."

(line 110) We rephrased the second main contribution of the study as: "...(ii) by using publicly open datasets we ensure the generalizability of the method, especially for DEM-related applications in datascarce regions;"

(line 117) Thank you for the comments on revising the overall significance of the study from "offering an exemplary pathway to address the issue of lacking high-resolution DEM for reliable risk assessments in the context of land use planning and disaster management" to focus on emphasising "improve flood simulation". However, given that we intend to keep the section on pluvial flood simulation evaluation in the main manuscript, we would like to argue that the section on quantifying the improvements in pluvial flood simulation also indicates the potential of improving broader applications across various domains that rely on high-resolution DEMs for reliable spatial analysis. Therefore, we think the original statement reflects the broader significance of the study, showcasing its value beyond flood simulations and positioning it as a methodological advancement applicable to multiple disciplines.

(line 120) We thank you for the comment about adding sentences guiding the reader into the method section. Accordingly, we added a guiding sentence as follows: "To improve the spatial resolution of DEM data for enhancing flood simulations, we further develop a deep learning-based DEM super-resolution method. This method employs the Residual Channel Attention Network (RCAN) (Y.Zhang et al., 2018) as the backbone structure and incorporates a tailored multi-source input block to leverage multi-sourced input data, contributing to improved performance in reconstructing high-resolution DEM data."

(line 135) As suggested, we also modified the last sentence of this paragraph to clarify our modifications on the backbone structure as follows: "However, since RCAN has been developed for image super-resolution tasks on single natural colour images, we tailored the structure of its input module to handle inputs from different data sources."

(line 140) As suggested, we modified section 2.2 "The input module enables the integration of highresolution multispectral satellite images with low-resolution DEM data, leveraging the complementary information from both sources to reconstruct high-resolution DEMs with enhanced accuracy (Fig. 6). Multispectral satellite images contain information captured across various spectral bands, including both visible light and near-infrared bands, which offer a wealth of information about surface materials, vegetation coverage, water bodies, and other landscape features (Carrão et al., 2008), making them ideal for compensating for the coarse information in low-resolution DEMs. By combining highresolution multispectral imagery with low-resolution elevation data, deep learning models can access a more comprehensive feature set, facilitating the reconstruction of detailed topographic information." (line 146) Thank you for the suggestion, we revised Figure 1 to emphasise the location of the multisource input module in the model structure, which now corresponds to the text: "The tailored multisource input module is integrated into the model structure before the first layer of the RCAN backbone structure (Fig. 1)."

---

## Author Response (AR2)

**Editor decision**

Dear Yue Zhu, Paolo Burlando, Puay Yok Tan, Christian Geiß, and Simone Fatichi,

Thank you for submitting your manuscript on 'Improving Pluvial Flood Simulations with Multisource DEM Super-Resolution' and for answering the reviewer's comments.

Two reviewers have examined the changes. As you can see from their comments, one is not fully convinced about the revised version.

Some issues remain, which I kindly ask you to take into consideration. Among other things, this refers to i) clarification of the scope of your study, ii) additional explanation of the simulation experiment.

As a next step, please respond to the reviewer's comments and upload the revised, marked-up (track changes) manuscript version. Additionally, you have to upload a clean new version of the manuscript.

If you have any questions regarding the procedure or interpretation of the comments, please do not hesitate to contact me for clarification.

Please also note that this decision does not necessarily imply acceptance of the manuscript in the journal NHESS. It depends on your edits to the manuscript based on the referees' comments and the editor's comments on the revised version.

I look forward to receiving the revised version of your manuscript.

Best regards Kai Schröter

Dear Prof Schröter,

We sincerely appreciate the time and effort the reviewers and editor invested in evaluating our manuscript and providing insightful comments. We have carefully considered each comment and have made corresponding revisions to the manuscript, including i) clarification of the scope of our study, ii) additional explanation of the experiment, iii) adding a README file to the openly shared dataset and codes, and iv) correction of Figure 6.

To facilitate the review process, we are including a detailed point-by-point response to the comments, indicating how each issue has been addressed. In the below point-by-point response, our responses are in *italic*, and the text in the revised manuscript is marked with quotation marks. Changes made to the manuscript are highlighted in the track-changes file.

We believe that these revisions have further enhanced our manuscript and hope that it will be considered favourably for publication. Thank you for considering our revised submission. Please do not hesitate to contact me if further information is needed.

Sincerely, Yue Zhu

**Report #2**

Thank you for your revision. The manuscript has been substantially improved, and the figures are much better now (with the exception of Fig. 6). However, most of my comments have not been adequately addressed and still stand:

With respect to the relevance to NHESS and the overall structure of the paper: I'm unconvinced that NHESS is the best journal for your work. (Yes, you have found—and now included—some more NHESS references; however, these are not DEM super-resolution focused, and clearly these were not foundational to your work.) Is the paper focused on improving DEM super-resolution? Then clearly it should be in a more ML-focused journal. Or is the paper about how simple DEM performance metrics fail to capture pluvial model performance? If so, then multiple pluvial models should be considered and a broader set of DEMs should be used—and probably an article like this would be better suited to a journal like HESS. Both of these would be interesting articles. However, combining these topics into a single article as you have confuses things.

We thank the reviewer for the thoughtful comment. We appreciate the opportunity to clarify the scope of this manuscript, and to explain why we believe NHESS is a very fit and impactful venue for this work.

While this study investigated machine learning methods for enhancing the spatial resolution of DEM data, its primary objective is to demonstrate how these methods can improve the resolution of terrain characterisation to a level, which allows to enhance significantly the accuracy of flood simulation. In this sense, the study sits at the intersection of natural hazard modelling – pluvial flood in the specific case – and processing of geospatial data needed for natural hazard assessment. We believe this aligns very well with the interdisciplinary scope of NHESS journal, which prominently includes the enhancement of modelling and assessment of natural hazards. Although the NHESS references we added were not purely focused on DEM super-resolution, they explore and highlight the influence of input data characteristics on hazard assessment outcomes, thus making the case for the foundational scope of this manuscript, which, ultimately, aligns well with the scope of NHESS.

Therefore, we believe this integration is actually a strength of the manuscript, because it demonstrates how the different performance of DEM super-resolution models can lead to differentiated improvements of their real-world hazard modelling (in this case hydrological) applications. We firmly believe that this has both scientific and applied relevance, because, to the best of our knowledge, the extent to which the quality of DEM spatial resolution – and its enhancement – contribute to the accuracy of pluvial flood simulation lacks quantification in the existing literature. It seems to us that this question must be highly relevant to the scientific community working on earth system science and natural hazards. Accordingly, to better clarify the scope of this study, we revised the manuscript as follows:

(line 105) "[...], this study aims to investigate the effectiveness and quantification of how pluvial flood simulations can be improved by using a deep learning-based DEM super-resolution construction method, which incorporates multispectral imagery, including the near-infrared band, as additional input. Accordingly, we develop an integrated methodological framework that allows for enhancing input data quality for practical improvements in flood simulation performance, specifically quantifying the extent to which the proposed method of DEM resolution enhancement can contribute to improved pluvial flood hazard simulations."

Regarding the use of a single pluvial model, our intention is, coherently with the scope of the article, not to benchmark flood simulation models, but to compare under controlled simulation settings the effect of different DEM enhancement techniques based on super-resolution methods. The employed flood simulation model is widely established in urban flood risk analysis, as put in evidence by the cited literature, so that we consider it to be suitable for representing a reference model on which to test how input data quality – the DEM resolution – affects its performance. Please also note that in reference to the hydraulic part, most urban flood models are solving either directly or indirectly some form 2D shallow water equations, and differences are often more related to the numerical implementation rather than to fundamental process description. Therefore, as far as the selected model is representative of this model category, it is likely to provide similar sensitivity to DEM as other urban flood models. Including multiple flood models, as suggested, would indeed broaden the scope of the research and enhance the generalisability of the results. However, it would also dilute the main message of this study and generate a significantly different study that goes beyond the scope of this article, as this study showcases for the first time the impact of different hyper-resolution DEM quality on pluvial flood simulations. The inclusion of additional pluvial flood models would be better suited for a follow-up study that focuses, for instance, on the relative performance of different flood simulation models based on a given DEM input quality. However, to account for this suggestion, we added a new point in the discussion section as follows:

(line 390) "[...], further tests could assess the effects of super-resolution DEM under alternative rainfall scenarios and using additional flood simulation models to assess if DEM input quality has some level of model dependency. While many flood models share the same fundamental equations to solve for flow processes (Guo, Guan, & Yu, 2021), such extended analyses would likely broaden the scope of the study and enhance the generalisability of the results."

- Thank you for sharing the code. I looked briefly at the code, and there is no README. This does not facilitate review or reuse and is unsatisfactory in my view. Also, there are many unnecessary cache files that have been included. What dependency versions were used?

Thank you very much for this comment. We updated the dataset by removing unnecessary cache files and adding a README file with information on dependency versions. The updated dataset can be accessed following this link: https://zenodo.org/records/15212783

I'm still unclear on how the simulation experiment was divided between the two study areas. Were they each trained separately (i.e., separate model weights)? If so, this implies the proposed method only works when the input 10 m DEM is already available (for the training)? What would the practical applications of something like this be? Instead, I think it would be more useful to train on one study area and apply these weights/inference on the other dataset.

Thank you for this comment. We would like to clarify the rationale of the experiment setup, also providing more details of the experiment. The two study areas were trained and tested independently on two different datasets, correspondingly having different model weights. Training and validating models in different datasets is regarded as a widely accepted experiment setup in many deep learning applications for remote sensing products (Huang, Zhao, & Song, 2018; Mou, Ghamisi, & Zhu, 2017; Shao, Zhou, Deng, Zhang, & Cheng, 2020; Yang et al., 2018). We chose this experiment setting because the goal of this study is to demonstrate method robustness across different contexts, not transferability, and evaluate method performance within each context to validate if it works properly under different geographical conditions, allowing the assessment of whether the proposed method could effectively learn terrain-specific representations that improve flood simulation accuracy.

While we acknowledge that training on one area and testing on another may provide insights into cross-site generalisation, such an approach without any adaptation is known to perform ineffectively, particularly in spatial learning tasks where input distributions differ significantly (Tuia, Persello, & Bruzzone, 2016). In this study, the two datasets vary in terms of acquisition sensors and terrain morphology. Applying a model trained on one directly to the other would introduce domain shift, leading to degraded performance and potentially misleading conclusions about model robustness. To clarify the settings of the training process on the two datasets, we made the following modifications in the manuscript:

(*line 205*) "All the models were trained and validated separately in the UK and China datasets, facilitating the evaluation of model performance in learning terrain-specific representations across different geographical contexts."

Also, the objective of our study is not to develop a universal or domain-invariant model, but to evaluate the utility of super-resolved DEMs in improving flood simulation accuracy within a given area. Demonstrating that this is the case on two significantly different terrain and urban landscapes is in its own a result, which has practical implications in real-world applications: for instance, limited high-resolution DEM data can be partially available for specific regions, and this framework demonstrates that a subset of such data is sufficient to train effective models locally. We agree that model transferability and generalisation are promising areas for future exploration, and this is very likely achievable by training the model with diverse global DEM datasets. We now highlight this in the discussion section as a direction for further research, particularly with pretraining on diverse global DEM datasets and fine-tuning them for local applications, as follows:

(line 385) "Moreover, this study trained and evaluated models separately for two different regions characterised by different geographical and terrain contexts. Future work, such as pretraining on diverse global DEM datasets and fine-tuning for specific local applications, could explore approaches to improve model generalizability, thus supporting broader applicability in regions with limited high-resolution DEM data."

- Guo, K., Guan, M., & Yu, D. (2021). Urban surface water flood modelling a comprehensive review of current models and future challenges. *Hydrology and Earth System Sciences*, *25*(5), 2843–2860. doi: 10.5194/hess-25-2843-2021
- Huang, B., Zhao, B., & Song, Y. (2018). Urban land-use mapping using a deep convolutional neural network with high spatial resolution multispectral remote sensing imagery. *Remote Sensing of Environment*, 214, 73–86. doi: 10.1016/j.rse.2018.04.050

- Mou, L., Ghamisi, P., & Zhu, X. X. (2017). Deep Recurrent Neural Networks for Hyperspectral Image Classification. *IEEE Transactions on Geoscience and Remote Sensing*, *55*(7), 3639–3655. doi: 10.1109/TGRS.2016.2636241
- Shao, Z., Zhou, W., Deng, X., Zhang, M., & Cheng, Q. (2020). Multilabel Remote Sensing Image Retrieval Based on Fully Convolutional Network. *IEEE Journal of Selected Topics in Applied Earth Observations and Remote Sensing*, 13, 318–328. doi: 10.1109/JSTARS.2019.2961634
- Tuia, D., Persello, C., & Bruzzone, L. (2016). Domain Adaptation for the Classification of Remote Sensing Data: An Overview of Recent Advances. *IEEE Geoscience and Remote Sensing Magazine*, 4(2), 41–57. doi: 10.1109/MGRS.2016.2548504
- Yang, Y., Dong, J., Sun, X., Lima, E., Mu, Q., & Wang, X. (2018). A CFCC-LSTM Model for Sea Surface Temperature Prediction. *IEEE Geoscience and Remote Sensing Letters*, 15(2), 207–211. doi: 10.1109/LGRS.2017.2780843
  - It seems Fig 6 was mistakenly replaced with content from Fig 5 during the revision.

Thank you very much for pointing this out. We have updated Fig 6 with the correct figure.